# Epidemiological Investigation and Characterization of Avian Influenza A *H3N8* Virus in Guangdong Province, China

**DOI:** 10.3390/ani15233377

**Published:** 2025-11-21

**Authors:** Junjie Lin, Yuze Li, Haojian Luo, Yiqiao Wang, Yingying Liu, Kun Mei, Feng Wen, Zhaoping Liang, Shujian Huang

**Affiliations:** 1School of Animal Science and Technology, Foshan University, Foshan 528231, China; 15918059606@163.com (J.L.); lyz19992023@163.com (Y.L.); 18664820794@163.com (H.L.); 13686930259@163.com (Y.W.); yyliu0603@163.com (Y.L.); kunmei@fosu.edu.cn (K.M.); 2College of Veterinary Medicine, South China Agricultural University, Guangzhou 510462, China; liangzp@scau.edu.cn

**Keywords:** *H3N8* AIV, serological survey, phylogenetic analysis, pathogenicity

## Abstract

This study investigates the epidemiological and virological features of the *H3N8* low pathogenic avian influenza virus in Guangdong Province, revealing key mutations, shifts in receptor specificity, and providing insights into its evolutionary path and zoonotic potential.

## 1. Introduction

Avian influenza viruses (AIVs), belonging to the genus Alphainfluenzavirus (Influenza A virus) of the Orthomyxoviridae family, exhibit significant host diversity, infecting wild *aquatic birds*, *humans*, *pigs*, *dogs*, and *horses* [1]. Among AIVs, *H3*-subtype influenza viruses represent a globally prevalent and evolutionarily dynamic group, phylogenetically divided into three main lineages based on host specialization: avian *H3Nx* (encompassing multiple neuraminidase (*NA*) subtypes such as *H3N2*, *H3N8*, and *H3N6*), seasonal human *H3N2*, and equine *H3N8* viruses [2]. Notably, *H3*-subtype viruses display substantial virulence variation across host species: most avian *H3Nx* viruses are low pathogenic avian influenza viruses (LPAIVs) in wild *waterfowl* (e.g., *mallards*) and poultry, but some can evolve into highly pathogenic avian influenza viruses (HPAIVs) via adaptive mutations or gene reassortment; seasonal human *H3N2* viruses typically cause mild to moderate respiratory illness in *humans*, while equine *H3N8* viruses induce severe respiratory disease in *horses* [2,3]. Key determinants of host specificity and pathogenicity for *H3*-subtype viruses primarily involve the hemagglutinin (*HA*) protein—especially amino acid mutations in its receptor-binding domain (RBD), such as Q226L and G228S, which shift binding affinity between SAα-2,3-Gal and SAα-2,6-Gal [2,4]. Influenza A viruses are classified into 19 *HA* and 11 *NA* subtypes based on the antigenic diversity of *HA* and *NA*, from which only two subtypes have non-avian hosts (*H17N10* and *H19N11*, *bats*) [5,6]. The spread of infectious diseases driven by global human interconnectedness has led to multiple pandemics over the previous century and past decade, with avian influenza being a case in point [7]. In 1968, the pandemic was caused by a new reassortment of the influenza A virus *H3N2* carrying two segments (*HA* and *PB1*) from *H3* avian viruses [8]. Descendants of that pandemic continue to circulate among *humans* as a seasonal flu viruses to date. Recently, *H3N8* subtype avian influenza viruses have breached the species barrier, causing human infections and fatalities. The Chinese CDC reported three confirmed cases of human *H3N8* virus infection with severe respiratory symptoms in 2022–2023, all linked to live poultry exposure. The *H3* subtype AIVs are constantly evolving, exhibiting frequent reassortment and cross-species transmission [9]. For example, first discovered in 1963, *H3N8* equine influenza viruses have recently impacted Asia and the Middle East [10]. Surveillance data indicate increasing *H3N8* avian virus isolations annually, with *H3N8* avian viruses prevalent throughout China and becoming one of the most frequently isolated AIV subtypes [8,11].

The genome of all influenza A viruses, including *H3N8*, comprises 8 single-stranded, negative-sense RNA segments, totaling approximately 13.6 kb, encoding at least 10 functional proteins: the RNA polymerase complex (*PB2*, *PB1*, *PA*), surface glycoproteins *HA* and *NA*, nucleocapsid protein (*NP*), matrix proteins (*M1*, *M2*), and non-structural proteins (*NS1*, *NS2*) [12]. Alterations in glycosylation sites of the *HA* protein, particularly those proximal to the receptor-binding domain (RBD), can significantly impact its binding affinity and specificity to host cell surface sialylated glycoprotein receptors by inducing steric hindrance or conformational changes, thereby modulating viral replication and host adaptation [13,14]. The frequent genome reassortment and antigenic drift of *HA* and *NA* drive continuous zoonotic outbreaks. Epidemiological studies on the *H3N8* virus indicate that its sporadic human infections exhibit certain observable epidemiological features, though these are based on a limited number of documented cases [15], and while the *HA* protein originates from AIVs, its affinity for SAα-2,6-Gal is significantly increased. This evolutionary adaptation suggests that changes in viral receptor-binding properties are a key driver in the emergence of influenza pandemics [16,17].

## 2. Materials and Methods

### 2.1. Ethics Statement

This study was approved by the Ethics Committee of the School of Animal Science and Technology, Foshan University (protocol code 20221818). All *chickens* were humanely handled in accordance with the protocols and principles of animal ethics. The veterinarians obtained written consent from the owners to collect samples.

### 2.2. Serological Survey of Poultry for the H3N8 Virus

A total of 2040 *chicken* and 677 *duck* serum samples were obtained from animal disease prevention and control centers across various prefecture-level cities in Guangdong Province (Table 1). Serum samples were collected in October 2022. Notably, these birds did not exhibit any clinical symptoms of influenza infection at the time of sampling. Serum samples were treated with Receptor Destroying Enzyme (RDE) at a 1:3 ratio and incubated at 37 °C for 18–24 h to remove non-specific inhibitors. RDE was subsequently inactivated by incubation in a 56 °C water bath for 30 min. Following cooling to room temperature, samples were diluted in phosphate-buffered saline (PBS) and serially two-fold diluted in 96-well plates. Hemagglutination inhibition (HI) assays were performed by adding 0.5% *chicken* red blood cells and 8 hemagglutinating units of antigen to each well, and the strain *A/chicken/Qingyuan/22/2022 (H3N8)* was used as antigen. After a 30 min incubation at 37 °C, HI titers were determined by visual inspection [18].

### 2.3. Case Sample Collection

From March to November 2022, pharyngeal swab samples were collected from diseased *chicken* flocks on a farm in Qingyuan City, Guangdong Province, and stored at the Veterinary Preventive Medicine Laboratory of Foshan University. Upon arrival at the laboratory, samples were mixed with phosphate-buffered saline (PBS, pH 7.4) containing 1% penicillin-streptomycin, subjected to three freeze–thaw cycles, and centrifuged at 1200 rpm for 5 min. Following sterile filtration, the supernatant was collected and stored for subsequent analysis.

### 2.4. Pathogen Isolation and Plaque Purification

Embryonated *chicken* eggs and *MDCK* cells were used to isolate viruses. 0-day-old *specific-pathogen-free* (*SPF*) *chicken embryos*, purchased from Xinxing Dahuainong Poultry and Egg Co., Ltd., were placed in an incubator at 37 °C, and their status was observed regularly daily. 10-day-old *chicken embryos* were selected for allantoic cavity inoculation with the sterile-filtered supernatant described in 2.3, with an inoculation dose of 200 μL per embryo. After inoculation, the inoculation sites were sealed, and the embryos were labeled with time and serial numbers before being placed in the incubator for continuous culture for 72 h. The condition of the *chicken embryos* was observed every 8 h. The allantoic fluid of embryos that died between 24 and 72 h was collected and labeled as the F1 generation. Hemagglutination assays (HAA) were performed on the collected allantoic fluid; those with hemagglutination activity were passaged continuously to the F3 generation. For each passage, the pathological changes and mortality of the *chicken embryos* were observed, and the allantoic fluid was stored in a −80 °C refrigerator for later use.

*Madin-Darby Canine Kidney* (*MDCK*) cells were cultured until they formed a dense monolayer. The culture medium was then removed, and the cells were washed twice with PBS. The treated samples were diluted in minimal essential medium (MEM), and 1 mL of the diluted sample was inoculated into a T25 culture flask (catalog number: 13112A, manufactured by Beijing Labselect Technology Co., Ltd., Beijing, China). After a 1 h incubation at 37 °C, 4 mL of MEM supplemented with 1 μg/mL TPCK-treated trypsin was added for further culture. Virus culture was harvested when the cytopathic effect (CPE) reached ≥75%, subjected to three freeze–thaw cycles, and passaged continuously until a stable CPE was observed. The resulting viral strain was then isolated and subjected to virus identification.

For plaque purification, the virus solution was serially diluted and inoculated onto *MDCK* cells. Following incubation, cells were overlaid with a low melting point agarose mixture (concentration: 2%; Cat. No. A8350; Manufacturer: Solarbio, Beijing, China). After 3–5 days of culture, individual plaques were selected, used to inoculate fresh *MDCK* cells, and the virus was amplified. This plaque purification process was repeated three times to obtain a purified viral strain. Viral nucleic acid was then extracted using a kit from Bioer Technology Co., Ltd., of Hangzhou, China, and the virus was identified.

To determine *HAA* titer, the purified virus fluid was serially diluted in 96-well V-bottom plates, and 0.5% *chicken* red blood cells were added. After a 30 min incubation at 37 °C, the HAA titer was determined by visual inspection, based on the highest dilution at which complete agglutination was observed [19].

### 2.5. Receptor-Binding Assays

Streptavidin-coated 96-well plates were used to bind varying concentrations of glycopolymers SAα-2,3-Gal and SAα-2,6-Gal (3′SLN-C3-BP, 6′SLN-C3-PAA-biot, GlycoNZ, Auckland, New Zealand), with three technical replicates per concentration and one negative control group. Plates were blocked with 5% bovine serum albumin (BSA) in PBS. Viral samples were diluted to a concentration of 16 hemagglutinating units (HAU) per 50 μL and added to the wells as antigens. Following incubation, viral nucleic acids were extracted from each well.

A SYBR Green-based Real-Time Quantitative PCR (RT-qPCR) assay was established for the detection of the *H3N8* virus using TB Green^®^ Premix Ex Taq™ II (Tli RNaseH Plus) from Takara Bio Inc. (Catalog No.: RR820A; location: Kusatsu, Shiga, Japan) on the qTOWER3 G instrument manufactured by Analytik Jena. The standard curve equation was Y = −3.72lgX + 42.52 (R^2^ = 0.997, E = 79.61%). The PCR conditions were initial denaturation at 94 °C for 3 min, followed by 40 cycles of denaturation at 94 °C for 30 s, annealing at 55 °C for 30 s, and extension at 72 °C for 20 s (primer sequences are listed in Table 2).

### 2.6. Complete Genome Sequencing

*H3N8* virus gene sequences published in GenBank were aligned and analyzed using MEGA 5.0 software. Premier 6.0 software was then used to design primers for whole-genome amplification. All primers were synthesized by Sangon Biotech (Shanghai) Co., Ltd. (primer sequences listed in Table 3). RNA was extracted using the RNAfast200 Total RNA Rapid Extraction Kit (Cat# 220011; Shanghai Feijie Biotechnology Co. Ltd., Shanghai, China). Then reverse transcription combined with PCR in one tube was performed using the HiScript II One Step RT-PCR Kit (Dye Plus) (Cat# P612-01; Vazyme Biotech Co., Ltd., Nanjing, China) with specific primers (see Table 3) in a thermocycler (qTOWER3 G, Analytik Jena, Jena, Thuringia, Germany) according to the following program: reverse transcription at 50 °C for 30 min, initial denaturation at 94 °C for 5 min, followed by 30 cycles of denaturation at 94 °C for 30 s, annealing at 55 °C for 30 s, and extension at 72 °C for 1 min/kb. The amplified products were purified via agarose gel electrophoresis and fragments of the expected size were sequenced by Sanger method using (BigDye™ Direct, Thermo Fisher) kit and sequence analyzer (Applied BiosystemsTM 3730XL, Thermo Fisher) (Sangon Biotech, Shanghai, China) [20]. The complete genome sequences of the three isolated *H3N8* strains have been deposited in the GenBank database. The GenBank accession numbers for QY15, QY21, and QY31 are PX454529-PX454536, PX454537-PX454544, and PX455047-PX455054, respectively.

### 2.7. Alignment and Phylogenetic Analysis

To elucidate the genetic and evolutionary relationships of the isolated strain, sequencing results were compared to published *H3N8* whole-genome sequences in GenBank using the Basic Local Alignment Search Tool for Nucleotides (BLASTn) algorithm at the National Center for Biotechnology Information (NCBI, available at https://www.ncbi.nlm.nih.gov/). Recombination analysis was performed by screening for sequences exhibiting the highest homology to each gene segment of the isolated strain. Phylogenetic trees for each gene segment were then constructed using the Maximum Likelihood method implemented in MEGA 5.0 software. The reliability of the phylogenetic trees was assessed using Bootstrap analysis with 1000 replicates.

### 2.8. Cleavage Site Analysis and Prediction of N-Glycosylation Sites

Potential N-glycosylation sites on the *HA* and *NA* proteins of the isolated strains were predicted using the NetNGlyc 1.0 server. The amino acid sequence at the *HA* protein cleavage site was also analyzed to predict the cleavage motif.

### 2.9. Pathological Experiments

To assess viral pathogenicity, 21-day-old *specific-pathogen-free* (*SPF*) *chickens* were randomly assigned to one of four groups. *chickens* in the treatment groups were inoculated with 100 µL of the QY22 strain (6.00 lg 50% Egg Infectious Dose (EID50)/mL) per bird via intranasal, oral gavage, and intramuscular routes, respectively. The control group received an equal volume of sterile PBS via the same routes. Clinical signs and weight changes were monitored daily in all groups for 10 consecutive days post-inoculation (dpi). Pharyngeal swabs were collected from all *chickens* at 1, 3, 5, 7, and 9 dpi to quantify viral shedding. At 3, 5, and 7 dpi, four *chickens* from each group were randomly selected and euthanized in accordance with approved animal care protocols. Euthanasia was performed via intravenous injection of sodium pentobarbital solution (dosage: 100–150 mg/kg body weight; concentration: 60 mg/mL). After injection, the *chickens*’ responses were observed: loss of consciousness and respiratory arrest occurred within 30–60 s. At 1–2 min post-injection, cardiac arrest was confirmed by palpation of the heart, and euthanasia was deemed successful. Necropsies were performed, and gross lesions were recorded. Samples of brain, nasal turbinate, trachea, lung, kidney, and bursa of Fabricius were collected for histopathology and viral load determination [21,22].

### 2.10. Histological Examination

Collected tissues and organs were fixed in 4% paraformaldehyde in PBS for 24 h at room temperature, followed by dehydration in a graded series of ethanol, clearing in xylene, and embedding in paraffin. Sections were cut at a thickness of 5 μm using a microtome. The sections were then stained with hematoxylin and eosin (H&E) and examined by light microscopy to assess pathological changes.

### 2.11. Viral Genome Quantity and Tissue Viral Load Detection

Nucleic acids were extracted from the collected pharyngeal/anal combined swabs and tissue/organ samples following three freeze-thaw cycles. The *H3N8* virus SYBR Green-based RT-qPCR assay, as described in Section 2.5 above, was used to quantify viral contents in the samples. Viral contents in pharyngeal/anal combined swabs and each organ were calculated based on the Quantification Cycle (Cq) values. These data were used to analyze Viral Genome Detection Dynamics and viral distribution characteristics in various organs at different time points post-inoculation.

## 3. Results

### 3.1. Prevalence of the H3N8 Virus in Guangdong Province, China

A serological survey for *H3N8* virus antibodies in Guangdong Province revealed an overall seroprevalence of 10.85% (221/2040) in *chicken* samples and 7.97% (54/677) in *duck* samples. Seroprevalence in *chicken* flocks was notably higher, with cities in the northeastern region of Guangdong Province, including Shaoguan, Meizhou, and Guangzhou, exhibiting seroprevalence rates exceeding 10% (Figure 1). Among *duck* flocks, Zhaoqing had the highest seroprevalence (11.76%), while Zhuhai had the lowest (3.22%) (Figure 2).

### 3.2. Isolation and Identification of Viruses

Three hemagglutinating agents, designated as QY15, QY22, and QY31, were obtained by both methods of virus isolation using *SPF chicken embryos* and *MDCK* cells. Infected *chicken embryos* had poor development and systemic hemorrhage in comparison with the control one (Figure 3). The death time of the *chicken embryos* was observed between 24 and 48 h.

The QY15, QY22, and QY31 were plaque-purified by infecting *MDCK* cells with serial 10-fold dilutions of the virus-positive tissue suspensions. Plaque formation was observed for all three isolates (Figure 4), with optimal plaque visibility achieved at a dilution of 10^−4^. The isolates were identified as *H3N8* subtype of influenza A virus based on sequencing results. Finally, they were named *A/chicken/Qingyuan/15/2022* (designated as QY15), *A/chicken/Qingyuan/22/2022* (QY22), and *A/chicken/Qingyuan/31/2022* (QY31). The strains agglutinated *chicken* red blood cells in the HAA with a titer of 64 HAU/50 μL for QY15, 256 HAU/50 μL for QY31, and 1024 HAU/50 μL for QY22.

### 3.3. Receptor Binding Profles of H3N8 Viruses

Detection of receptor-binding specificity of the isolated strains revealed that all three strains could bind to both SAα-2,3-Gal and SAα-2,6-Gal. Among them, the QY22 strain exhibited stronger binding affinity to SAα-2,3-Gal compared with the QY15 and QY31 strains, while its binding ability to SAα-2,6-Gal was weaker (Figure 5).

### 3.4. Genetic Analysis of the H3N8 Virus

Whole-genome phylogenetic analysis placed the *H3* gene within the Eurasian lineage and the *N8* gene within the North American lineage. The *H3* gene shares high homology with the Guangdong isolates *A/chicken/Guangzhou/199/2022(H3N8)*, *A/chicken/Dongguan/868/2022(H3N8)* and *A/chicken/Dongguan/364/2022(H3N8)*, while the *N8* gene was closely related to *A/chicken/Dongguan/879/2022(H3N8)* and *A/chicken/Guangzhou/4463/2021(H3N8)*. In addition, the internal genes were classified into the Y439-like lineage (*PB2*), G1-like lineage (M), and F98-like lineage (*PB1*, *PA*, *NP*, *NS*) (Figure 6) [3]. Among them, internal genes such as *PB1*, *PA*, and *NP* clustered with *H9N2* viruses prevalent in Guangdong Province in recent years. Further analysis of these internal genes revealed that the *PA* genes of all three strains share a common ancestral origin with *chicken H9N2* viruses circulating in China in 2021, with *A/chicken/China/2096/2021(H9N2)* serving as a representative homologous strain. In the *NP* gene phylogenetic tree, QY15 and QY22 exhibit identical nucleotide sequences in their *NP* genes, which cluster with *chicken H9N2* viruses prevalent in South Korea and China between 2020 and 2022; notably, all three *H3N8* strains harbor *NP* genes acquired from *H9N2* viruses. For the *M* and *NS* genes of the three isolates, they share a common precursor with those of other *chicken*-origin *H3N8* viruses circulating in Guangdong Province in 2022. These two genes are also present in some *chicken*-origin *H9N2* viruses and *duck*-origin *H3N8* viruses—specifically the *M* gene of *A/chicken/China/KM212/2022(H9N2)*, the *NS* gene of *A/chicken/China/XD3/2022(H9N2)*, and both the *M* and *NS* genes of *A/duck/Jiangxi/447/2022(H3N8)*. Additionally, the *PB1* genes of QY15, QY22, and QY31 share a common ancestor with that of *A/pigeon/Fujian/3.15_FZHX0008-C/2018(H9N2)*, a strain whose *PB1* gene is derived from early avian *H9N2* viruses. Phylogenetic analysis of complete genomes suggests that the three studied isolates are related to the *chicken*-origin, *duck*-origin, and *pigeon*-origin *H9N2* viruses present in China in recent years. Their genomes may have originated earlier through complex genetic reassortment events between *H9N2* and *H3N8* viruses, which have been circulating among different *poultry*, *waterfowl*, and *wild* or *migratory birds*.

In the *HA* receptor-binding site, these strains possess the amino acids 138A, 190E, 194L, 225G, 226Q, and 228G, which are highly conserved among avian *H3* viruses (Figure 7) [4]. However, the QY22 strain has D198, which differs from the A198 present in QY15 and QY31; the A198D substitution in the QY22 strain alters the charge and size of the amino acid at this position. Position 198 is adjacent to key sites such as position 190, and this may partially explain the differences observed in receptor-binding assays (Figure 5). This could account for the weak binding affinity of the QY22 strain to SAα-2,6-Gal.

When comparing the *HA* amino acid sequences of the *chicken* QY15, QY22, and QY31 isolates with those of the human *H3N8* virus *A/Changsha/1000/2022* (a human-adapted *H3N8* strain), it was found that these *chicken* isolates, like the human isolate *A/Changsha/1000/2022*, do not possess the key human adaptation-related amino acid substitutions (Q226L, G228S) and still maintain the avian signatures (Q226 and G228) in the 220-loop [2]. However, they share the same E190 residue—a charged amino acid that can synergize with sites such as Q226 and G228 to enhance binding affinity for SAα-2,6-Gal [4]. Additionally, within the receptor-binding site, both these *chicken* isolates and the human strain *A/Changsha/1000/2022* harbor the three amino acid residues N193, W222, and S227 (Table 4). Among these, the N193 residue has the potential to form hydrogen bonds with α2-6-linked glycans, while the W222 and S227 residues may alter the conformational flexibility of the 220-loop [2]. These two effects collectively endow the QY15, QY22, and QY31 *H3N8* isolates with dual receptor-binding properties.

### 3.5. Cleavage Site and Predicted N-Glycosylation Sites

Analysis of the *HA* protein cleavage site of the three isolated strains revealed the motif PEKQTR↓GLF, which contains two basic amino acid residues (K at position 342 and R at position 345 according to *HA* precursor numbering). The three strains shared the same predicted N-glycosylation sites on the *HA* and *NA* proteins (Figure 8), consistent with those observed in *H3N8* viruses prevalent in Shantou, Dongguan, Guangzhou, and Huizhou in Guangdong Province in 2022.

### 3.6. Pathological Analysis

Following virus inoculation, body weight and cloacal temperature changes in *SPF chickens* were monitored daily for 10 consecutive days post-inoculation (Figure 9). Weight gain rates in the inoculated groups began to decline from 4 dpi, with *chickens* inoculated via the intranasal route exhibiting the lowest weight gain. Cloacal temperatures fluctuated within the normal range of 40.5–41.8 °C. Lethargy was observed from 5 dpi in inoculated groups. No mortality was recorded during the experiment. Necropsy (Figure 10) revealed gross lesions including cerebral hemorrhage, laryngeal swelling, tracheal hemorrhage, and pulmonary congestion in inoculated *chickens*. Lesions in other organs were minimal. *chickens* inoculated via the intranasal route exhibited the most pronounced hemorrhage in various organs.

### 3.7. Histological Analysis

For the intranasal inoculation group, the brain exhibited perivascular cuffing, the nasal turbinate exhibited mucosal epithelial necrosis, the trachea exhibited submucosal lymphocytic infiltration, the lung displayed alveolar edema, the spleen presented lymphoid depletion, and the bursa of Fabricius showed follicular atrophy. In the oral gavage group, mild tracheal epithelial damage and pulmonary interstitial inflammation were observed. The intramuscular injection group exhibited slight splenic lymphoid hyperplasia. Conversely, the control group had normal tissue architecture in all organs. Collectively, *chickens* in the inoculated groups showed more severe lesions than those in the control group (Figure 11).

### 3.8. Viral Genome Detection Dynamics

To investigate the dynamics of viral genome fragments at different time points, pharyngeal/anal combined swabs were collected from 0 to 11 dpi. Viral genome was detectable in the nasal and oral groups as early as 1 dpi, with peak levels at 5 dpi (nasal: 10^5.7^ copies/μL; oral: 10^2.8^ copies/μL), then decreased gradually. In contrast, no viral genome was detected in the intramuscular group during the entire observation period. These results indicate that nasal inoculation is more conducive to viral genome replication and persistence in the host (Figure 12).

### 3.9. Analysis of Tissue Viral Load

To determine viral loads at various time points, Viral load analysis of tissues and organs at 3, 5, and 7 dpi demonstrated that the QY22 strain has broad tissue tropism, with viral shedding in the upper (nasal turbinate, trachea) and lower respiratory tracts (lung). Notably, the nasal group exhibited the highest viral loads in the nasal turbinate (10^5.8^ copies/μL), brain (10^5.1^ copies/μL), trachea (10^4.9^ copies/μL), lung (10^4.7^ copies/μL), bursa of Fabricius (10^4.7^ copies/μL), and kidney (10^4^·^5^ copies/μL) (Figure 13).

## 4. Discussion

As a major *waterfowl*-breeding country, China accounts for three-quarters of the global breeding volume. Domestic *ducks* are predominantly raised in southern regions through high-density and free-range farming [23,24]. The *H3Nx* virus has the highest isolation rate in *ducks*, reaching 91.76% [25]. Previous serological surveys also highlighted the prevalence of *H3* subtype avian influenza viruses. From 2006 to 2007, Pu et al. [26] reported an overall positive rate of 2.83% for *H3Nx* in 173 *chicken* flocks across 18 provinces in China, with notably higher rates in the southeastern coastal provinces of Jiangsu and Guangdong. More recently, in 2018–2019, Liu et al. [27] conducted testing on 2474 poultry samples from Jiangsu Province, revealing a 1.57% positive rate for the *H3Nx* virus. Within these positive samples, *chickens* exhibited a 12.47% positive rate, and *ducks* showed 19.15%.

Building upon these historical trends, our current study, conducted in 2022, investigated the seroprevalence of *H3N8* virus antibodies in *chicken* and *duck* sera from various cities in Guangdong Province. Among 2040 *chicken* serum samples collected, the overall positive rate was 10.85%. For the 677 *duck* serum samples, the overall positive rate stood at 7.97%. The seroprevalence of *H3N8* in *chickens* remains comparatively high, with cities exceeding a 10% positive rate concentrated in northeastern Guangdong Province (e.g., Shaoguan, Meizhou, and Guangzhou). These findings collectively suggest that *H3N8* virus infection rates in both *chickens* and *ducks* in Guangdong Province remain elevated. This sustained high infection rate is particularly noteworthy given that *ducks* often serve as asymptomatic carriers or exhibit only mild symptoms upon infection, while *chickens*, being a primary commodity in live poultry markets, appear to demonstrate a greater susceptibility to the virus, leading to higher observed infection rates.

In this study, three *H3N8* viruses, designated QY15, QY22, and QY31, were isolated and investigated. The *HA* gene of these isolates belonged to the Eurasian lineage, while the *NA* gene was of North American origin. Furthermore, phylogenetic analysis of the internal genes revealed their classification into distinct lineages: *PB2* belonged to the Y439-like lineage, *M* to the G1-like lineage, and *PB1*, *PA*, *NP*, and *NS* to the F98-like lineage [3]. Notably, several internal genes, including *PB1*, PA, and *NP*, clustered with *H9N2* viruses that have been prevalent in Guangdong Province in recent years. Among these, the *PA* genes of all three strains share a common ancestral origin with *chicken H9N2* viruses circulating in China in 2021 (with *A/chicken/China/2096/2021(H9N2)* as a representative homologous strain). With respect to the *M* and *NS* genes of the three isolates, they share a common precursor with those of other *chicken*-origin *H3N8* viruses prevalent in Guangdong Province in 2022. These two genes are also present in some *chicken*-origin *H9N2* viruses and *duck*-origin *H3N8* viruses—specifically the *M* gene of *A/chicken/China/KM212/2022(H9N2)*, the *NS* gene of *A/chicken/China/XD3/2022(H9N2)*, and both the *M* and *NS* genes of *A/duck/Jiangxi/447/2022(H3N8)*. The *NP* genes of QY15 and QY22 exhibit identical nucleotide sequences and cluster with *chicken H9N2* viruses prevalent in South Korea and China during 2020–2022, and all three *H3N8* strains harbor *NP* genes acquired from *H9N2* viruses. Additionally, the *PB1* genes of QY15, QY22, and QY31 share a common ancestor with that of the *pigeon*-origin *H9N2* virus *A/pigeon/Fujian/3.15_FZHX0008-C/2018(H9N2)*, whose *PB1* gene is derived from early avian *H9N2* viruses. This suggests that the *H9N2* gene module may have long served as an “internal gene donor”, promoting the adaptability of recombinant viruses [28,29]. Notably, the three strains carry internal genes derived from *H9N2*, and their genomes may have formed earlier through complex genetic reassortment events between *H3N8* viruses and *H9N2* viruses circulating among different poultry, *waterfowl*, and wild or *migratory birds*. This highlights the crucial role of the *waterfowl*-land*fowl* ecological interface in gene exchange, consistent with the viral diversity hotspot characteristics observed in the migratory bird habitats of the Pearl River Delta wetlands [30]. Research has demonstrated that influenza A viruses (IAVs), including *H3N8* viruses, can persist in aquatic environments for extensively long periods under favorable conditions—for instance, Ramey et al. [31] found that IAVs remained infectious for more than seven months in surface waters of northern wetlands in North America. These wetlands were identified as biologically important media for viral transmission and maintenance, even serving as environmental reservoirs for IAVs during the overwintering period of *migratory birds*. This gene reassortment event significantly increases the likelihood of the virus acquiring new traits, thereby enhancing its transmission potential and pathogenicity [25].

Analysis of the *HA* receptor-binding site revealed that the QY15, QY22, and QY31 isolates collectively retain the typical avian-signature residues (Q226 and G228). This indicates that these *H3N8* isolates have not yet acquired the core genetic markers for efficient adaptation to the human host, which is a key factor limiting their direct and widespread transmission among *humans*. Second, the E190 residue in the three isolates can synergize with Q226 and G228 to enhance the virus’s binding affinity for SAα-2,6-Gal [4]. This common genetic feature suggests that despite retaining avian signatures, the *H3N8* isolates may have gained a certain capacity to bind to human-type receptors, laying a potential foundation for cross-species infection. Furthermore, the N193 residue of the isolates has the potential to form hydrogen bonds with α2-6-linked glycans, which may further strengthen the virus’s binding to human cells; the W222 and S227 residues of the isolates may alter the conformational flexibility of the 220-loop—a region critical for receptor recognition, indirectly adjusting the virus’s ability to interact with different types of sialic acid receptors [2]. Collectively, the synergistic effects of the above amino acid residues endow the QY15, QY22, and QY31 *H3N8 chicken* isolates with dual receptor-binding properties. This dual-binding capability is a key intermediate phenotype in the process of influenza virus adaptation from avian to human hosts. QY15, QY22, and QY31 isolates exhibit a preference for binding to SAα-2,6-Gal but retain the ability to bind SAα-2,3-Gal. Consistent with the receptor-binding properties of the *chicken*-origin *H3N8* viruses [*A/chicken/Fujian/F0101/2022(H3N8)*, *A/chicken/Henan/F0308/2022(H3N8)*, *A/chicken/Anhui/FE12/2022(H3N8)*, *A/chicken/Jiangsu/B314/2022(H3N8)*] isolated in 2022 by Yang et al. [2] and the *duck*-origin *H3N8* viruses [*A/duck/Guangxi/446D28/2021(H3N8)*] isolated in 2024 by Li et al. [20]. These *H3N8* virus isolates exhibit the ability to bind to and potentially infect human receptors, posing a risk of crossing the species barrier to infect mammals and *humans*.

*SPF chickens* were infected with the purified QY22 strain to assess its pathogenicity. The virus demonstrated efficient replication in the *chicken* respiratory system, with lung tissue viral titers reaching 5.04 lg EID50/mL. This viral load was significantly higher than those previously observed for both the four *duck*-origin *H3N8* viruses [*A/duck/Guangdong/S1286/2009(H3N8)*, *A/duck/Anhui/S4053/2010(H3N8)*, *A/duck/Jiangxi/S3855/2012(H3N8)*, *A/duck/Zhejiang/S4088/2013(H3N8)*] isolated in 2016 by Cui et al. [11] and the three *chicken*-origin *H3N8* isolates, which were isolated in 2022 by Yang et al. [2]. Furthermore, the infection resulted in organ damage beyond the respiratory tract, including the brain, kidneys, and bursa of Fabricius. These findings indicated that, in comparison to earlier *waterfowl*-origin *H3N8* viruses, the novel *H3N8* viruses harboring the internal genes of *H9N2* exhibited elevated pathogenicity in *chickens*. This suggests that the acquisition of *H9N2* internal genes is a key factor contributing to the increased infection and pathogenicity observed in *chickens*.

## 5. Conclusions

This study systematically investigated the epidemiological characteristics, genetic evolution patterns, and virological properties of the avian influenza A *H3N8* virus circulating in Guangdong Province, China, providing comprehensive evidence for an in-depth understanding of the virus’s epidemic dynamics and zoonotic risk. Serological surveillance of 2040 *chicken* serum samples and 677 *duck* serum samples from 21 prefecture-level cities in Guangdong Province in 2022 revealed an overall seroprevalence of 10.85% in *chickens* and 7.97% in *ducks*. Notably, the seroprevalence in *chickens* from northeastern Guangdong was significantly higher, indicating that the *H3N8* virus persists and spreads widely in local avian populations, with *chickens* being more susceptible to the virus than *ducks*. Three *H3N8* virus strains (QY15, QY22, QY31) were successfully isolated from diseased *chicken* flocks in Qingyuan City, and these viruses were confirmed to possess dual receptor-binding specificity—a key phenotypic adaptive trait that facilitates the virus’s potential cross-species transmission.

Genetic and phylogenetic analyses uncovered the complex evolutionary characteristics of the isolates: the *HA* gene clustered within the Eurasian lineage, while the NA gene belonged to the North American lineage. The internal genes were classified into the Y439-like lineage (*PB2*), G1-like lineage (*M*), and F98-like lineage (*PB1*, *PA*, *NP*, *NS*). Among them, internal genes such as *PB1*, *PA*, and *NP* clustered with *H9N2* viruses prevalent in Guangdong Province in recent years. Additionally, the *pigeon*-origin *H3N8* strain *A/pigeon/Fujian/3.15_FZHX0008-C/2018(H9N2)* shared a common ancestor with the *PB1* gene of the isolates, and the *duck*-origin *H3N8* virus *A/duck/Jiangxi/447/2022(H3N8)* shared a common precursor for the *M* and *NS* genes with the three isolates. These findings indicate that the three *H3N8* isolates originated from genetic reassortment between *H3N8* and *H9N2* viruses circulating among *poultry*, *waterfowl*, and migratory bird-related populations, highlighting the role of the Pearl River Delta wetland ecosystem as a hotspot for viral gene exchange. Analysis of the amino acid sequence of the *HA* receptor-binding domain revealed the presence of both conserved avian-signature residues (Q226, G228) and adaptive mutation sites (E190, N193, W222, S227). These mutations, in synergy with conserved residues, enhance the virus’s binding affinity for human-type receptors while retaining its tropism for avian-type receptors, laying a molecular foundation for the virus’s zoonotic potential.

Pathogenicity assessment in the *SPF chicken* model demonstrated that the QY22 strain replicates efficiently in the respiratory tract and exhibits broad tissue tropism, with high viral loads detected in tissues such as the nasal turbinate, brain, trachea, lung, and bursa of Fabricius. *chickens* inoculated intranasally showed the most severe pathological lesions, including cerebral hemorrhage, tracheal hemorrhage, and alveolar edema, along with the highest viral genome load level, which peaked at 5 dpi post-inoculation (10^5.7^ copies/μL). Compared with early *waterfowl*-origin *H3N8* viruses, the acquisition of internal genes from *H9N2* viruses is associated with increased pathogenicity of this strain in *chickens*, underscoring the impact of genetic reassortment on viral adaptability. Our study contributes to a better understanding of the evolutionary trajectory and zoonotic potential of the *H3N8* low pathogenic avian influenza virus circulating in Guangdong and emphasizes the importance of strengthening surveillance of the *H3N8* virus in poultry populations to mitigate the risk of cross-species transmission and prevent potential public health threats.

## Figures and Tables

**Figure 1 animals-15-03377-f001:**
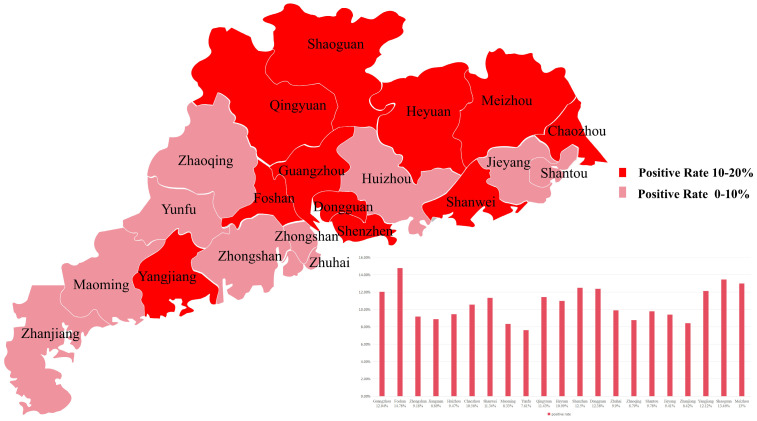
Spatial distribution of *H3N8* antibody positive rates in 2040 *chicken* serum samples collected from 21 prefecture-level cities in Guangdong Province of China during 2022. Seroprevalence was categorized into two ranges: 10–20% (red) and 0–10% (pink). Cities in northeastern Guangdong (e.g., Shaoguan, Meizhou, and Guangzhou) exhibited higher seroprevalence (>10%), while other regions showed relatively lower rates.

**Figure 2 animals-15-03377-f002:**
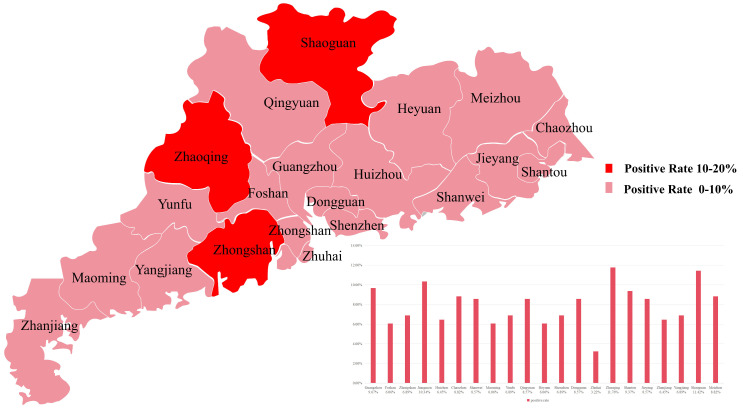
Spatial distribution of *H3N8* antibody positive rates in 677 *duck* serum samples collected from 21 prefecture-level cities in Guangdong Province of China during 2022. Seroprevalence was categorized into two ranges: 10–20% (red) and 0–10% (pink). Zhaoqing showed the highest seroprevalence (11.76%), while Zhuhai exhibited the lowest (3.22%).

**Figure 3 animals-15-03377-f003:**
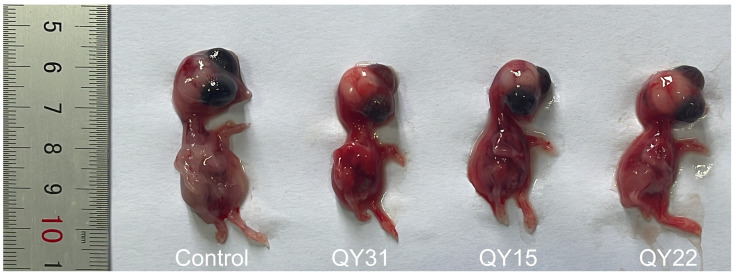
From left to right, they are the control group, the group inoculated with QY31 strain, the group inoculated with QY15 strain, and the group inoculated with QY22 strain. The *chicken embryos* shown in the photo are 11 days old, and the period post infection is 36 h. All the *chicken embryos* were isolated at the same period post infection.

**Figure 4 animals-15-03377-f004:**
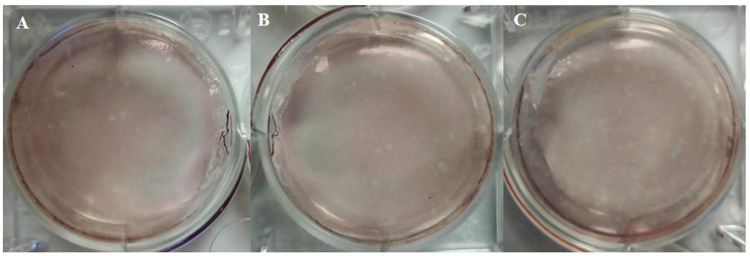
Plaque morphologies of three *H3N8* strains (QY15, QY22, QY31) during the small plaque selection process. Standard plaque assays were performed using *MDCK* cells, with the cells coated with 2% agarose, followed by incubation for 3–5 days. Viral plaques appear as white round spots. (**A**): QY15; (**B**): QY22; (**C**): QY31.

**Figure 5 animals-15-03377-f005:**
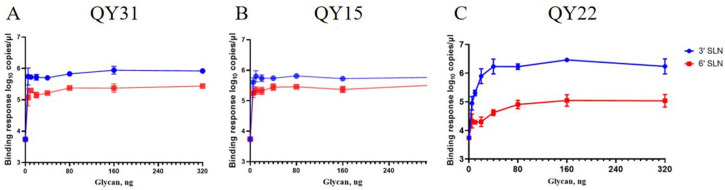
Receptor-binding specificity of HA from *H3N8* strains (QY31, QY15, QY22) to sialic acid receptors. (**A**): QY31 strain: Dual binding to both receptors with preferential binding to 3′ SLN, with moderate affinity for SAα-2,6-Gal; (**B**): QY15 strain: Dual binding to both receptors, with slightly higher affinity for SAα-2,3-Gal; (**C**): QY22 strain: Strongest binding to SAα-2,3-Gal, with weaker but detectable affinity for SAα-2,6-Gal. Data represent mean values from three technical replicates, indicating strain-specific receptor tropism variations.

**Figure 6 animals-15-03377-f006:**
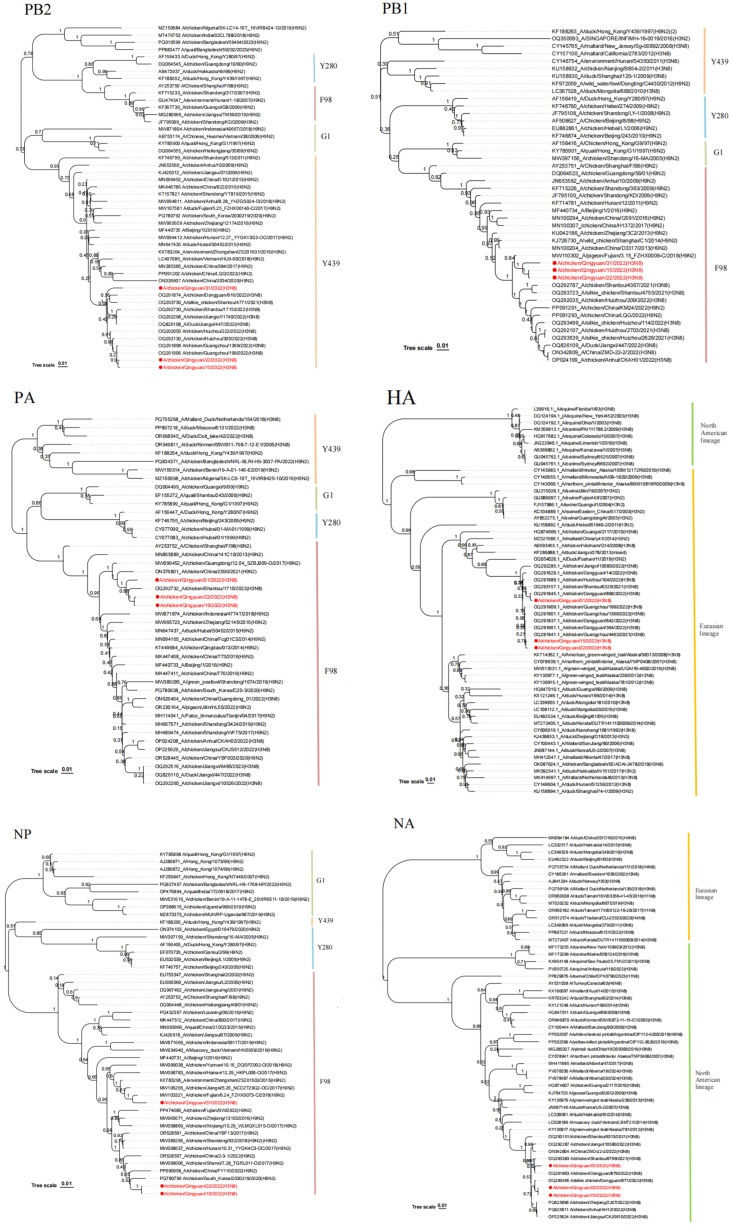
Phylogenetic analysis of internal gene segments (*PB2*, *PB1*, *PA*, *NP*, *M*, *NS*) of the *H3N8* isolates. The *PB2* gene clustered within the Y439-like lineage, the *M* gene within the G1-like lineage, and *PB1*, *PA*, *NP*, and *NS* genes within the F98-like lineage. Isolates are indicated by red font in the phylogenetic tree. These clusters indicate genetic reassortment events with *H9N2* viruses prevalent in Guangdong Province, highlighting the contribution of internal gene modules to viral evolution.

**Figure 7 animals-15-03377-f007:**
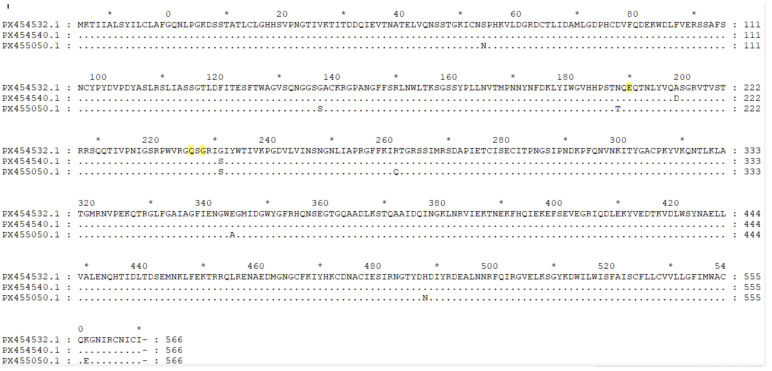
PX454532.1: QY15 strain; PX454540.1: QY22 strain; PX455050.1: QY31 strain. In the *HA* receptor-binding site, these strains possess the amino acids 138A, 190E, 194L, 225G, 226Q, and 228G that are highly conserved among avian *H3* viruses. Note that the symbol * denotes round tens (i.e., numbers that are multiples of ten), and the three positions highlighted in yellow represent the 190E, 226Q, and 228G sites, respectively. However, the A198D substitution in the QY22 strain alters the charge and size of the amino acid at this position, which may partially explain the differences observed in receptor-binding assays (Figure 5).

**Figure 8 animals-15-03377-f008:**
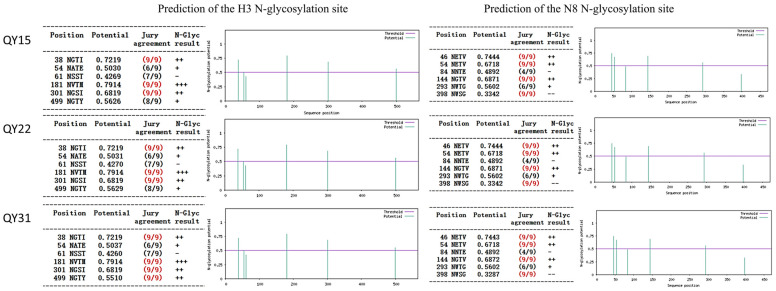
Predicted N-glycosylation sites in the *HA* and *NA* proteins of *H3N8* strains (QY15, QY22, QY31). Data were obtained due to the NetNGlyc 1.0 server. Symbol definitions for N-glycosylation prediction results: −, No predicted N-glycosylation site; +, Weakly predicted N-glycosylation site; ++, Moderately predicted N-glycosylation site; +++, Strongly predicted N-glycosylation site. For *H3* glycosylation sites: Conserved sites at positions 38 (NGTI), 181 (NVTM), and 301 (NGSI) were identified in all strains. For *N8* glycosylation sites: Conserved sites at 46 (NETV), 54 (NETV), 144 (NGTV), and 398 (NWSG) were detected in all strains. These results indicate their potential role in viral antigenicity and host adaptation.

**Figure 9 animals-15-03377-f009:**
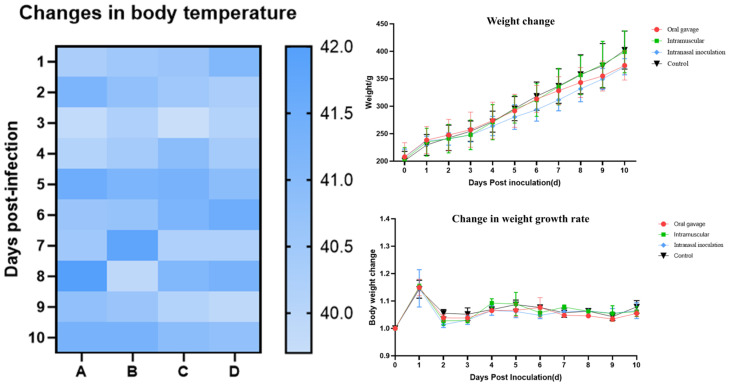
Temperature, body weight, and change in weight growth rate in *SPF chickens* after inoculation with the QY22 *H3N8* strain. (A): Intranasal inoculation group; (B): Oral gavage group; (C): Intramuscular injection group; (D): Control group (sterile PBS). Cloacal temperatures remained within the normal range (40.5–41.8 °C) throughout the observation period. Weight gain rates in inoculated groups declined from 4 dpi, with the intranasal group showing the lowest weight gain. Data were plotted using GraphPad Prism 8 and represent mean ± SD, with significant differences in weight gain observed from 5 dpi between inoculated and control groups, *p* < 0.05.

**Figure 10 animals-15-03377-f010:**
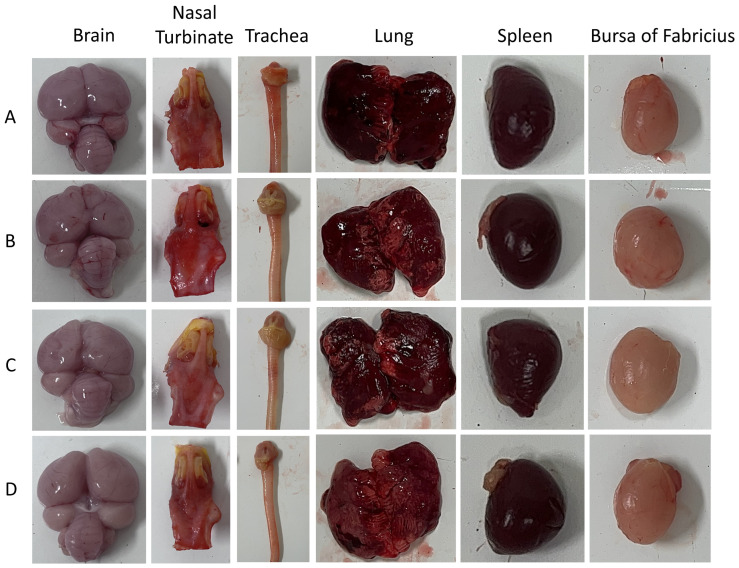
Gross lesions of *SPF chickens* infected with QY22 *H3N8* strain at autopsy. (**A**): Intranasal inoculation group with pronounced cerebral hemorrhage, laryngeal swelling, tracheal hemorrhage, and pulmonary congestion; (**B**): Oral gavage group showing mild tracheal congestion and pulmonary edema; (**C**): Intramuscular injection group exhibiting mild pulmonary congestion; (**D**): Control group with no visible lesions in brain, nasal turbinate, trachea, lung, spleen, or bursa of Fabriciu. Lesions were most severe in the intranasal group, consistent with efficient respiratory tract infection.

**Figure 11 animals-15-03377-f011:**
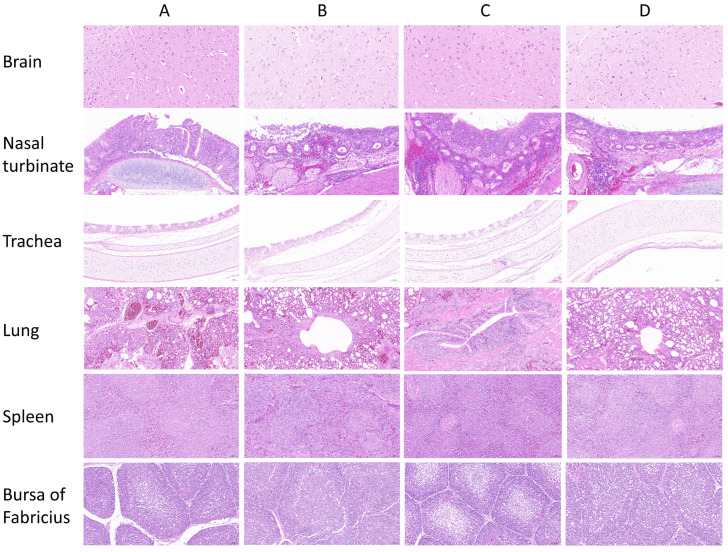
Histopathological images of *SPF chickens* tissue sections stained with hematoxylin and eosin (H&E) (magnification, ×200). The scale bar included in each image presents a length of 100 μm. (**A**): Intranasal inoculation group with brain showed perivascular cuffing, nasal turbinate exhibited mucosal epithelial necrosis, trachea had submucosal lymphocytic infiltration, lung displayed alveolar edema, spleen showed lymphoid depletion, bursa of Fabricius had follicular atrophy; (**B**): Oral gavage group showing mild tracheal epithelial damage and pulmonary interstitial inflammation; (**C**): Intramuscular injection group exhibiting slight splenic lymphoid hyperplasia; (**D**): Control group with normal tissue architecture in all organs.

**Figure 12 animals-15-03377-f012:**
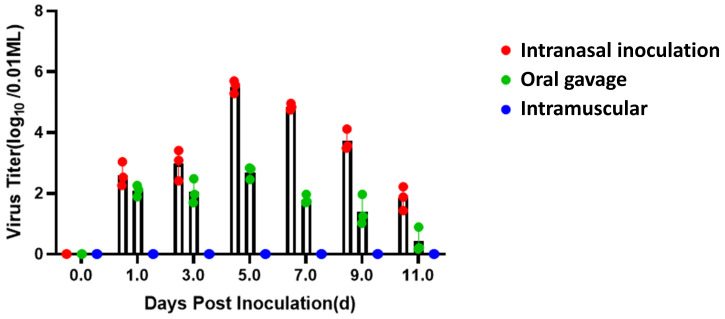
The dynamics of viral genome detection. Intranasal group: detectable viral genome from 1 dpi, peaking at 5 dpi (10^5.7^ copies/μL), here, 10^5.7^ copies/μL represents a logarithmic expression of the viral genome copy number per microliter, which is mathematically equivalent to 10^5^ × 10^0.7^ copies/μL (approximately 5.01 × 10^5^ copies/μL); oral gavage group: lower peak levels (10^2.8^ copies/μL at 5 dpi), similarly, corresponding to 10^2^ × 10^0.8^ copies/μL (approximately 6.31 × 10^2^ copies/μL); intramuscular group: no detectable viral genome. Note: The viral genome copy numbers (i.e., 10^5.7^ copies/μL and 10^2.8^ copies/μL) were calculated based on the standard curve established in Section 2.5. pharyngeal/anal swabs are expressed as mean ± SD, with significant differences between intranasal and other groups at 5 dpi, *p* < 0.01.

**Figure 13 animals-15-03377-f013:**
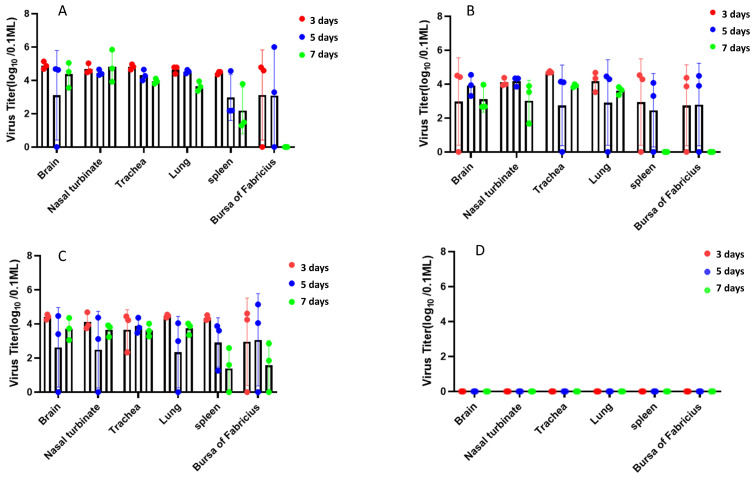
Analysis of viral load in organs of *SPF chicken*. (**A**): the nasal group exhibited the highest viral loads in the nasal turbinate (10^5.8^ copies/μL), brain (10^5.1^ copies/μL), trachea (10^4.9^ copies/μL), lung (10^4.7^ copies/μL), bursa of Fabricius (10^4.7^ copies/μL), and kidney (10^4.5^ copies/μL); (**B**): the Oral gavage group exhibited low viral loads in the spleen and the bursa of Fabricius at 7 dpi.; (**C**): the intramuscular injection group had low viral loads in the spleen and the bursa of Fabricius at 7 dpi.; (**D**): the Control group showed no detectable virus in any organ. Viral loads are expressed as mean ± SD, with significant differences between intranasal and other groups at 3 dpi, *p* < 0.01.

**Table 1 animals-15-03377-t001:** Number of *chicken* and *duck* serum samples tested in Guangdong province.

Prefecture-Level City	Number of *chicken* Serum Samples	Number of *duck* Serum Samples	Prefecture-Level City	Number of *chicken* Serum Samples	Number of *duck* Serum Samples
Guangzhou	108	31	Shenzhen	96	29
Foshan	115	33	Dongguan	105	35
Zhongshan	98	29	Zhuhai	101	31
Jiangmen	90	29	Zhaoqing	91	34
Huizhou	95	31	Shantou	92	32
Chaozhou	85	34	Jieyang	85	35
Shanwei	97	35	Zhanjiang	95	31
Maoming	96	33	Yangjiang	99	29
Yunfu	92	29	Shaoguan	104	35
Qingyuan	105	35	Meizhou	100	34
Heyuan	91	33	Total	2040	677

**Table 2 animals-15-03377-t002:** Identification primers of RT-qPCR.

Primers	Sequences (5′-3′)	Product Size (bp)
*H3*-F	GACCTGACCGACTCTGAG	91
*H3*-R	GAAGCAACCATTGCCCATA	

**Table 3 animals-15-03377-t003:** Identification primers of PCR.

Primers	Primer Sequences (5′-3′)	Product Size (bp)
*PB2*-F	TAYGARGARTTCACAATGGT	2370
*PB2*-R	TCYTCYTGTGARAAYACCAT	
*PB1*-F	ARATACCNGCAGARATGCT	2370
*PB1*-R	TTRAACATGCCCATCATCAT	
*PA*-F	CATTGAGGGCAAGCTTTC	2262
*PA*-R	TNGTYCTRCAYTTGCTTATCAT	
*HA*-F	CCATGAAGACTATCATTGCTTTGAGC	1765
*HA*-R	GCACTCAAATGCAAATGTTGCACCTAATG	
*NP*-F	TAGGAGCAAAAGCAGGGTA	1594
*NP*-R	TGGAGTAGAAACAAGGGTATTTTT	
*NA*-F	ATGGAGCAAAAGCAGGA	1489
*NA*-R	GGCCAGTAGAAACAAGGAGTTTTTT	
*M*-F	ACGGAGCAAAAGCAGGTAG	1056
*M*-R	CGGAGTAGAAACAAGGTAGTTTTT	
*NS*-F	TGGAAGCAAAAGCAGGGTG	919
*NS*-R	TGGAGTAGAAACAAGGGTGTTTT	

**Table 4 animals-15-03377-t004:** Comparison of key amino acids in the *HA* receptor-binding site between *H3N8* virus isolates (QY15, QY22, QY31) and the human isolate *A/Changsha/1000/2022*.

Mature HA Amino Acid Positions	QY15 Strain	QY22 Strain	QY31 Strain	*A/Changsha/1000/2022*
190	E	E	E	E
193	L	L	L	L
222	W	W	W	W
226	Q	Q	Q	Q
227	S	S	S	S
228	G	G	G	G

## Data Availability

The data generated in this study are available from the corresponding author upon reasonable request.

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
