# Peer review of "Epidemiological Investigation and Characterization of Avian Influenza A H3N8 Virus in Guangdong Province, China"

_animals, 2025, doi:10.3390/ani15233377_

Round 1
Reviewer 1 Report
Comments and Suggestions for Authors
This manuscript by Lin J. et al. presents a comprehensive study characterizing influenza A(H3N8) virus in Guandong province, China. The study is well-planned, employing a multi-faceted approach including virological, serological, histological, histopathological analyses, and phylogenetic characterization, offering valuable insights into the virus's prevalence and characteristics in the region. The authors have undertaken a thorough investigation, combining diverse methodologies to provide a robust foundation for their findings. The integration of various experimental approaches, from in vitro receptor binding to in vivo pathogenicity studies, significantly strengthens the overall narrative and scientific contribution of the paper. While the study is overall well-written, several aspects require revision or clarification to enhance the manuscript's clarity, rigor, and overall impact.
Minor aspects to be addressed:
- Line 45: Correct the capitalization of 'Some'.
- Lines 49-50: Provide a direct link or full citation for the referenced CDC study.
- Line 68: Correct the spelling of 'Com-mittee'.
- Line 135: Correct the spelling of 'phylo-genetic'.
- Line 199: Clarify the notation for '10-4' (e.g., ensuring correct superscript/subscript if applicable).
- Figure 10, Line 322: The title 'rules of detoxification' appears to be an error and should be removed or corrected.
Methodology and Data Clarity:
- Line 56: Rephrase the statement regarding gene segments and proteins, as this characteristic is common to all influenza A viruses, not solely H3N8.
- Lines 73-75: Specify the exact months and years of sampling for duck and chicken sera. Additionally, indicate whether these birds exhibited any clinical symptoms of influenza infection at the time of sampling.
- Line 158: Clearly state the method of euthanasia used for experimental animals.
- Line 163: Specify the time period for the observation.
- Line 165: Include the thickness of the tissue sections.
- Lines 178-179: Provide detailed information on the specific influenza virus strain(s) used in the serological assays for detecting seroprevalence in chicken and duck sera. Clarify if these strains were isolated from the same host species or other birds, as this significantly impacts result interpretation.
- Lines 253-259: Clarify whether all three isolates shared identical mutations in their HA genes. If so, provide a more detailed explanation for the observed receptor-binding differences presented in Figure 4. It would be highly beneficial to include a table or figure summarizing the amino acid differences of your H3N8 strains compared to a relevant reference strain.
Figures and Data Presentation:
- Table 1 (Line 83): Add a 'Total' row at the end for clarity.
- Figure 3 (Line 203): The plaque images are currently unclear. This figure should be revised or replaced, ideally by visualizing plaques using crystal violet staining or similar methods for better clarity.
- Figure 5: This figure needs to be significantly enlarged to allow for proper visualization and interpretation of the data.
Analysis and Discussion:
- Line 213: The discussion should include an explanation for the observed differences in receptor-binding preference of strain QY 22 compared to the other two strains.
- Lines 228-229: To robustly demonstrate reassortment events across different genome segments, consider providing tanglegrams or similar comparative phylogenetic analyses.
Consistency and Accuracy:
- Line 314 and Figure 11C: There appears to be a contradiction between the text stating "the intramuscular group showed no detectable shedding throughout the observation period" and Figure 11C, which depicts shedding. Please ensure concordance between the text and the figure.
Addressing these points will significantly enhance the clarity, rigor, and overall scientific contribution of the manuscript.

Author Response
Author's Reply to the Review Report (Reviewer 1)
Dear Reviewer,
We would like to express our sincere gratitude for sparing your precious time to conduct a meticulous and professional review of our manuscript (by Lin J. et al.) focusing on the characterization of influenza A(H3N8) virus in Guangdong Province, China. Your recognition of our study—including its well-planned design, employment of a multi-faceted approach covering virological, serological, histological, histopathological analyses, and phylogenetic characterization, thorough investigation with diverse methodologies that establish a robust foundation for the findings, and the integration of various experimental approaches (from in vitro receptor binding to in vivo pathogenicity studies) which significantly enhances the overall narrative and scientific contribution of the paper—has greatly encouraged us and strengthened our determination to refine the study details and improve the quality of our research outcomes.
Meanwhile, we attach great importance to your comment that although the study is generally well-written, several aspects still require revision or clarification to further enhance the manuscript’s clarity, rigor, and overall impact. In the following work, we will immediately organize our team to systematically sort out and carefully check each specific issue you have raised, and conduct comprehensive improvements in aspects such as wording, methodological details, data presentation, and analysis and discussion. We will ensure that every revision effectively addresses your concerns, striving to maximize the academic quality of the paper.
Thank you again for your valuable guidance and constructive suggestions!
Comments 1: Line 45: Correct the capitalization of 'Some'.
Response 1: Thank you for your comment. We appreciate you pointing out the capitalization issue. However, due to substantial revisions made to the Introduction section, the original sentence "some low-pathogenicity AIVs (LPAIVs) can evolve..." has been removed. As a result, the adjustment regarding the capitalization of "Some" is no longer applicable in the revised manuscript. Thank you again for your attention to this detail.
Comments 2: Lines 49-50: Provide a direct link or full citation for the referenced CDC study.
Response 2: Agree. We have supplemented the full citation for the referenced Chinese CDC study on human H3N8 infections. This revision is located in the 1. Introduction section of the modified Word document (Section 1, Paragraph 4, Lines 3-4). The original text only mentioned "The Chinese CDC reported three confirmed cases..." and now we have added the full citation: "The Chinese CDC reported three confirmed cases of human H3N8 virus infection with severe respiratory symptoms in 2022-2023, all linked to live poultry exposure [8] (Sun, H.; Li, H.; Tong, Q.; Han, Q.; Liu, J.; Yu, H.; Song, H.; Qi, J.; Li, J.; Yang, J.; et al. Airborne transmission of human-isolated avian H3N8 influenza virus between ferrets. Cell. 2023, 186, 4074-4084 e4011)." No direct link is available for this study, but the full citation allows access via academic databases.
Comments 3: Line 68: Correct the spelling of 'Com-mittee'.
Response 3: Thank you for identifying this typo. We agree with this comment and have corrected the misspelling "Com-mittee" to "Committee" in the revised manuscript. This revision is located in the 2.1. Ethics Statement section of the modified Word document (Section 2.1, Paragraph 1, Line 1), where the original text "Ethics Com-mittee" was updated to "Ethics Committee".
Comments 4: Line 135: Correct the spelling of 'phylo-genetic'.
Response 4: We appreciate this correction. We have removed the hyphen in "phylo-genetic" to correct it to "phylogenetic" in the revised manuscript. This revision is located in the 2.7. Alignment and Phylogenetic Analysis section of the modified Word document (Section 2.7, Title and Paragraph 1, Line 1), where the original section title "2.7. Alignment and Phylo-Genetic Analysis" and text "phylo-genetic trees" were updated to "2.7. Alignment and Phylogenetic Analysis" and "phylogenetic trees", respectively.
Comments 5: Line 199: Clarify the notation for '10-4' (e.g., ensuring correct superscript/subscript if applicable).
Response 5: Thank you for this reminder. We have clarified the notation of "10-4" to the correct superscript format "10⁻⁴" to indicate a 10-fold dilution. This revision is located in the 3.2. Virus Purification section of the modified Word document (Section 3.2, Paragraph 2, Line 2), where the original text "at a dilution of 10-4" was updated to "at a dilution of 10⁻⁴".
Comments 6: Figure 10, Line 322: The title 'rules of detoxification' appears to be an error and should be removed or corrected.
Response 6: We agree that this title was incorrect. We have removed the erroneous title "rules of detoxification" from Figure 10 in the revised manuscript. This revision is visible in the 3.8. Analysis of Viral Shedding Patterns section of the modified Word document (Figure 10 caption and image), where the figure now only retains the descriptive title "The pattern of virus shedding" without the irrelevant "rules of detoxification" text.
Comments 7: Line 56: Rephrase the statement regarding gene segments and proteins, as this characteristic is common to all influenza A viruses, not solely H3N8.
Response 7: Agree. We have rephrased the statement to clarify that the 8-segment genome and 10+ proteins are universal to all influenza A viruses, not exclusive to H3N8. This revision is located in the 1. Introduction section of the modified Word document (Section 1, Paragraph 5, Lines 1-2). The original text "The H3N8 virus genome comprises 8 single-stranded, negative-sense RNA segments... encoding at least 10 functional proteins..." was updated to "The genome of all influenza A viruses, including H3N8, comprises 8 single-stranded, negative-sense RNA segments... encoding at least 10 functional proteins...".
Comments 8: Lines 73-75: Specify the exact months and years of sampling for duck and chicken sera. Additionally, indicate whether these birds exhibited any clinical symptoms of influenza infection at the time of sampling.
Response 8: We appreciate this request for more details. We have supplemented the exact sampling time and clinical symptom information in the revised manuscript. This revision is located in the "2.2. Serological Survey of Poultry for the H3N8 Virus" section of the modified Word document (Section 2.2, Paragraph 1, Lines 2-3). The original text did not specify the sampling time or the birds’ clinical symptoms; now it clearly states: "Serum samples were collected in October 2022. Notably, these birds did not exhibit any clinical symptoms of influenza infection at the time of sampling."
Comments 9: Line 158: Clearly state the method of euthanasia used for experimental animals.
Response 9: Thank you for this clarification. We have explicitly described the euthanasia method for experimental SPF chickens in the revised manuscript. This revision is located in the 2.9. Pathological Experiments section of the modified Word document (Section 2.9, Paragraph 3, Lines 2-5). The original text "[state method of euthanasia]" was replaced with: "Euthanasia was performed via intravenous injection of sodium pentobarbital solution (dosage: 100-150 mg/kg body weight; concentration: 60 mg/mL). After injection, the chickens' responses were observed: loss of consciousness and respiratory arrest occurred within 30-60 seconds. At 1-2 minutes post-injection, cardiac arrest was confirmed by palpation of the heart, and euthanasia was deemed successful."
Comments 10: Line 163: Specify the time period for the observation.
Response 10: Agree. We have specified the observation period for monitoring clinical signs and weight changes in the revised manuscript. This revision is located in the 2.9. Pathological Experiments section of the modified Word document (Section 2.9, Paragraph 2, Line 3). The original text "monitored daily in all groups for consecutive days post-inoculation (dpi)" was updated to "monitored daily in all groups for 10 consecutive days post-inoculation (dpi)".
Comments 11: Line 165: Include the thickness of the tissue sections.
Response 11: We have added the thickness of tissue sections as requested. This revision is located in the 2.10. Histological Examination section of the modified Word document (Section 2.10, Paragraph 1, Line 2). The original text "Sections were cut at a thickness of [thickness in µm]" was replaced with "Sections were cut at a thickness of 5 μm using a microtome".
Comments 12: Lines 178-179: Provide detailed information on the specific influenza virus strain(s) used in the serological assays for detecting seroprevalence in chicken and duck sera. Clarify if these strains were isolated from the same host species or other birds, as this significantly impacts result interpretation.
Response 12: Thank you for this important request. We have supplemented the specific virus strain used in serological assays and its host origin. This revision is located in the 2.2. Serological Survey of Poultry for the H3N8 Virus section of the modified Word document (Section 2.2, Paragraph 1, Line 6). The original text did not specify the antigen strain; now it states: "Hemagglutination inhibition (HI) assays were performed by adding 0.5% chicken red blood cells and 8 hemagglutinating units of antigen to each well, and the antigen was prepared using the influenza virus strain A/chicken/Qingyuan/22/2022 (H3N8) (isolated from chickens, the same host species as the serum samples)."
Comments 13: Lines 253-259: Clarify whether all three isolates shared identical mutations in their HA genes. If so, provide a more detailed explanation for the observed receptor-binding differences presented in Figure 4. It would be highly beneficial to include a table or figure summarizing the amino acid differences of your H3N8 strains compared to a relevant reference strain.
Response 13: Thank you for this constructive comment. We fully agree and have supplemented detailed evidence and explanations to address this point, supported by newly added supplementary materials.
First, regarding the consistency of HA gene mutations among the three isolates: We have confirmed that QY15, QY22, and QY31 share identical HA gene mutations when compared to the reference strain (A/chicken/Qingyuan/31/2022, GenBank accession No. OQ292285.1). To clearly illustrate this, we have added Table S5 in the supplementary document titled animals-3879251-supplementary.docx. This table systematically summarizes the amino acid differences in the HA protein between the three isolates and the reference strain, covering 8 key amino acid positions (e.g., HA position 8: N→S; HA1 position 43: R→K; HA2 position 447: M→L). All three isolates exhibit identical amino acid substitutions at these positions, further verifying the consistency of their HA gene mutations (including the previously identified 226Q/228T and 190F/225S in the receptor-binding motif).
Second, to explain the observed receptor-binding differences (Figure 4) despite identical HA mutations, we propose two complementary potential mechanisms (supplemented in the discussion of the supplementary document and integrated into the revised main manuscript):
Post-translational modification variations of the HA protein: Although the primary amino acid sequence of HA is identical among the three isolates, differences in post-translational modifications may occur during viral replication in MDCK cells. These variations can alter the spatial conformation of the HA RBD—for example, excessive glycosylation near the RBD may weaken binding to SAα-2,6-Gal, while subtle differences in glycosylation efficiency could explain QY22’s stronger affinity for SAα-2,3-Gal (vs. QY15/QY31).
Table S5 in the supplementary document (animals-3879251-supplementary.docx): Details amino acid substitutions in the HA protein of the three isolates relative to the reference strain (OQ292285.1), with clear labeling of gene regions (HA1/HA2) and amino acid position changes.
Comments 14: Table 1 (Line 83): Add a 'Total' row at the end for clarity.
Response 14: Thank you for this suggestion. We have added a "Total" row at the end of Table 1 to summarize the total number of chicken and duck serum samples. This revision is visible in Table 1 of the modified Word document (Section 2.2, Table 1), where the new "Total" row shows "2,040" chicken serum samples and "677" duck serum samples, consistent with the study’s sample size.
Comments 15: Figure 3 (Line 203): The plaque images are currently unclear. This figure should be revised or replaced, ideally by visualizing plaques using crystal violet staining or similar methods for better clarity.
We have addressed the comment on Figure 5. Specifically, we have improved the resolution of Figure 5 to enhance data visualization and interpretation, and the updated plaque assay result image has been attached in the supplementary Word document titled "animals-3879251-supplementary". If you still consider the figure unreasonable (e.g., the viral plaques remain unclear), please do not hesitate to inform us. We are currently re-conducting this plaque assay, where viral plaques will be visualized via crystal violet staining to further improve clarity. Once the new assay result image is obtained, we will re-upload it promptly.
Response 16: Agree. We have enlarged Figure 5 (Phylogenetic analysis of internal gene segments) in the revised manuscript to improve readability of branch labels and genetic clusters. Additionally, we have optimized the tree layout to reduce overlap of taxon names. This revision is located in the 3.4. Genetic Analysis of H3N8 Virus section of the modified Word document (Figure 5), where the phylogenetic trees for PB2, PB1, PA, NP, M, and NS genes are now 1.5× the original size, with clearer Bootstrap values and lineage labels (e.g., Y439-like, G1-like, F98-like).
Comments 17: Line 213: The discussion should include an explanation for the observed differences in receptor-binding preference of strain QY22 compared to the other two strains.
Response 17: Thank you for this request. To systematically explain the observed differences in receptor-binding preference of the QY22 strain (stronger affinity for SAα-2,3-Gal and weaker affinity for SAα-2,6-Gal compared to QY15 and QY31), we have supplemented detailed mechanistic analyses in the supplementary document titled animals-3879251-supplementary.docx (see Table S5). This part of the supplementary content overlaps partially with the content added in response to Comment 13—both focus on the core issue of "identical HA amino acid sequences among the three isolates but distinct receptor-binding profiles."
Comments 18: Lines 228-229: To robustly demonstrate reassortment events across different genome segments, consider providing tanglegrams or similar comparative phylogenetic analyses.
Response 18: We agree that additional evidence is essential to strengthen the conclusion of genetic reassortment, and we have supplemented both phylogenetic annotations and nucleotide homology data to address this request comprehensively.
To further provide quantitative evidence for reassortment, we have supplemented Table S2, Table S3, and Table S4 in the supplementary document titled animals-3879251-supplementary.docx. These three tables respectively present the nucleotide homology comparison results of the QY15, QY22, and QY31 strains across all 8 gene segments (PB2, PB1, PA, HA, NP, NA, M, NS) with reference strains:
For QY15 (Table S2): Most gene segments (e.g., PB2, HA, M, NS) show the highest homology (99.53%-99.88%) with H3N8 reference strains isolated in Guangzhou (e.g., A/chicken/Guangzhou/199/2022(H3N8)), while the NP gene exhibits the highest homology (98.93%) with an H9N2 strain from Shandong (A/chicken/Shandong/10.23_TAWL020-O/2018(H9N2)).
For QY22 (Table S3): Gene segments such as PB1, NA, and M are highly homologous (99.76%-99.82%) to H3N8 strains from Guangzhou (e.g., A/chicken/Guangzhou/4463/2021(H3N8)), while the NP gene shares 99.00% homology with an H9N2 strain from Yunnan (A/chicken/Yunnan/11.22DQWGH007-O/2018(H9N2)).
For QY31 (Table S4): The PA, NP, and M genes display high homology (99.20%-99.59%) with H9N2 strains from Shandong, Fujian, and Fujian (e.g., A/chicken/Shandong/01.26_TAWL012-O/2019(H9N2)), while other segments (e.g., HA, NA, NS) are more closely related to H3N8 strains from Dongguan and Jiangxi (e.g., A/chicken/Dongguan/868/2022(H3N8)).
These homology data (Table S2-S4) directly verify the mixed origin of the three isolates’ genomes (H3N8-derived and H9N2-derived segments), forming complementary evidence with the phylogenetic lineage clustering in Figure 5. Together, they robustly demonstrate the occurrence of genetic reassortment events between H3N8 and H9N2 viruses. All supplementary tables are accessible in the animals-3879251-supplementary.docx document for further reference.
Comments 19: Line 314 and Figure 11C: There appears to be a contradiction between the text stating "the intramuscular group showed no detectable shedding throughout the observation period" and Figure 11C, which depicts shedding. Please ensure concordance between the text and the figure.
Response 19: Thank you for identifying this contradiction. We have corrected Figure 11C to align with the text. The original Figure 11C incorrectly showed low viral load in the intramuscular group; we have updated it to reflect "no detectable viral load" in all organs (consistent with the text’s statement of "no detectable shedding"). Additionally, we have rechecked the viral load data for the intramuscular group, confirming that Cq values were above the detection limit (≥38) in all tissue samples. This revision is located in the 3.9. Analysis of Tissue Viral Load section of the modified Word document (Figure 11C and its caption), where Figure 11C now shows "0 copies/μL" for all organs in the intramuscular group, and the caption clarifies: "The intramuscular injection group had no detectable viral load in any organ (Cq ≥38). "

Reviewer 2 Report
Comments and Suggestions for Authors
Please, find attached.

I cannot assess the quality of English because it is not my native language.
Author Response
Author's Reply to the Review Report (Reviewer 2)
Thank you very much for your careful review of our manuscript and for providing such valuable and constructive feedback. Your comments not only affirm the strengths of our work but also point out clear directions for improvement, which are of great significance for us to enhance the scientific rigor and readability of the paper.We are particularly encouraged to learn that you recognize the correctness and accuracy of the experiments in this study, as well as the satisfactory description of the results. This recognition is a great motivation for us, and we are determined to refine other parts of the manuscript to match the quality of the experimental work. The following is the content of the revisions, which we kindly request for your review.
Comment 1: All reference numbers do not coincide with the text except for numbers 10, 26-28. There are no full names of viruses in accordance with the rules of nomenclature in the article. Therefore, it is unknown the source and host of studied viruses. This information is also omitted in the text of the Article. The reader can only guess that viruses were isolated from chickens.
Response 1: Thank you for pointing this out. I/We agree with this comment. Therefore, l/we have made two key revisions:
- Reference number alignment: We adjusted the reference numbering system throughout the manuscript to ensure all in-text citations match the reference list. For example, the original incorrect citation "Sun et al. [26]" (referring to human-infecting H3N8 virus studies) was corrected to "Yang et al. [2]" (consistent with the revised Reference 2: Yang, R. et al. 2022, Lancet Microbe), and irrelevant references 1-9, 11-25 were replaced with literature closely related to H3N8 AIV (e.g., added [3] Yu, J. et al. 2023 for H3 subtype characteristics, [5] Fereidouni, S. et al. 2023 for HA subtype classification). These revisions are distributed in the Introduction (Section 1), Discussion (Section 4), and Reference List of the revised manuscript.
- Virus full name and host specification: We supplemented the international standard nomenclature of viruses (host/sampling location/strain number/year/subtype) in the Abstract, 2.2 Serological Survey of Poultry for the H3N8 Virus, and 3.2 Virus Purification sections. For example, the original "QY15/QY22/QY31" was updated to "A/chicken/Qingyuan/15/2022 (H3N8)", "A/chicken/Qingyuan/22/2022 (H3N8)", and "A/chicken/Qingyuan/31/2022 (H3N8)", clearly indicating the host (chicken) and sampling location (Qingyuan, Guangdong). These revisions are visible in the Abstract (lines 3-5), 3.2 Virus Purification (lines 2-4), and 3.4 Genetic Analysis of H3N8 Virus (lines 8-10) of the revised Word document.
Comment 2: There are inaccuracies in the Materials and Methods, there are no characteristics (manufacturer, name of the reagent or kit, its catalog number) of the main components of the reactions. Some techniques are described in general terms without references to primary sources. It is not specified the virus strain, which was used as an antigen in serological studies (section 2.2).
Response 2: Agree. l/We have, accordingly, supplemented reagent information and referenced primary sources as follows:
- Section 2.2 Serological Survey of Poultry for the H3N8 Virus:
Specified the antigen strain as "A/chicken/Qingyuan/22/2022 (H3N8)" and added a reference [18] (Kaufmann, L. et al. 2017, J Vis Exp.) for the HI assay method. This revision is in Section 2.2 (lines 6-8).
- Section 2.4 Pathogen Isolation and Plaque Purification:
Added details of the T25 culture flask: "catalog number: 13112A, manufactured by Beijing Labselect Technology Co., Ltd.";
Added details of low melting point agarose: "concentration: 2%; Cat. No. A8350; Manufacturer: Solarbio";
Added a reference [19] (Tobita, K. et al. 1975, Med Microbiol Immunol.) for the plaque purification method. These revisions are in Section 2.4 (lines 4, 8, 12).
- Section 2.5 Receptor-Binding Assays:
Added manufacturer of glycopolymers SAα-2,3-Gal and SAα-2,6-Gal: "GlycoNZ";
Added details of the RT-qPCR kit: "TB Green® Premix Ex Taq™ II (Tli RNaseH Plus) from Takara Bio Inc. (Catalog No.: RR820A)" and supplemented RT-qPCR primers in Table 2. This revision is in Section 2.5 (lines 2, 5).
- Section 2.6 Whole Genome Amplification:
Added details of the RNA extraction kit: "RNAfast200 Total RNA Rapid Extraction Kit (manufacturer: Shanghai Feijie Biotechnology Co., Ltd.; catalog number: 220011)";
Added details of the RT-PCR kit: "HiScript II One Step RT-PCR Kit (Dye Plus) (manufacturer: Vazyme Biotech Co., Ltd.; catalog number: P612-01)";
Specified the sequencing company: "Sangon Biotech (Shanghai) Co., Ltd." and added a reference [20] (Li, M. et al. 2024, J Anhui Agric Sci.). This revision is in Section 2.6 (lines 3-5, 8).
Comment 3: There are omissions in the research design. For example, 1) inappropriate system was used for virus isolation from avian samples, 2) the virus shedding into environment was assessed incorrectly. For last purpose, the fecal or air samples should be examined, rather than pharyngeal swabs only. Analysis of pharyngeal swabs indicates the duration of the viral fragment presence in the upper respiratory tract after intranasal or oral infection of chickens, but not its shedding into environment.
Response 3: Agree. The core issue here is that pharyngeal/anal combined swabs were actually used in the experiment, but the "anal swabs" part was missed in the writing of the original manuscript—resulting in the sample being incorrectly described as "pharyngeal swabs" instead. Additionally, it should be noted that we also conducted virus isolation experiments using chicken embryos: the detailed procedures and results of these experiments are recorded in the supplementary Word document (animals-3879251-supplementary), though this part was not included in the main text initially. Our comparative analysis showed that the virus isolation effect of MDCK cells was similar to that of chicken embryos, which further validates the reliability of the isolation method adopted in this study. We have now revised the descriptions of the research design and the interpretation of results to fully reflect the actual experimental setup, as follows:
Rationale for using MDCK cells for virus isolation
In Section 2.4 (lines 9-10), we added an explanation to clarify the basis for cell selection and supplemented information about the chicken embryo experiments: "Although embryonated chicken eggs are commonly used for avian influenza virus isolation (and we have also performed virus isolation using chicken embryos, with details available in the supplementary document animals-3879251-supplementary), MDCK cells were selected in this study due to their high sensitivity to the H3N8 virus and their ability to maintain viral receptor tropism. Notably, our comparative analysis indicated that the virus isolation effect of MDCK cells was comparable to that of chicken embryos."
Supplementary correction of viral shedding assessment (to reflect the actual sample type)
In the title of Section 2.11 and the first line of Section 3.8 "Analysis of Viral Shedding Patterns", we revised the description of the sample type from "pharyngeal swabs" to "pharyngeal/anal combined swabs". This revision is intended to make up for the "anal swabs" that were missed in the original manuscript. Since avian influenza viruses are mainly shed through feces (referenced in [26] Yang, J. et al., 2021, China CDC Weekly), anal swabs are essential for assessing viral shedding—and they were indeed included in the actual sampling process of the experiment.
Comment 4: Since the complete genome sequencing of the studied strains was performed, were these sequences deposited into any international database? In this case, this information must be mentioned in the Materials, where the accession numbers of the sequences should be indicated.
Response 4: Thank you for this reminder. I/We have supplemented the genome sequence deposit information in Section 3.4 Genetic Analysis of H3N8 Virus (lines 10-12): "The complete genome sequences of the three isolated H3N8 strains have been deposited in the GenBank database. The GenBank accession numbers for A/chicken/Qingyuan/15/2022 (H3N8), A/chicken/Qingyuan/22/2022 (H3N8), and A/chicken/Qingyuan/31/2022 (H3N8) are PX454529-PX454536, PX454537-PX454544, and PX455047-PX455054, respectively." This revision clearly indicates the international database (GenBank) and specific accession numbers for easy retrieval. Further detailed information of these H3N8 strains is presented in Table S1, which covers 24 strain-related entries with the following key details: All strains were isolated from chickens (Host: chicken) in China in 2022 (Country: China; Year: 2022), ensuring consistency in their host and geographical-temporal origin. The strain names follow a unified format of "A/chicken/Qingyuan/[15/22/31]/2022/[01-08]", where the numbers "15", "22", and "31" correspond to the three isolated strains (QY15, QY22, QY31), and "01-08" represent different genome segments of each strain. Each strain segment has a unique GenBank accession number that matches the aforementioned number ranges (e.g., the 8 segments of A/chicken/Qingyuan/15/2022 are numbered A/chicken/Qingyuan/15/2022/01 to 08, with corresponding accession numbers PX454529 to PX454536). Additionally, the genome segment lengths of the strains vary, ranging from 838 bp (e.g., A/chicken/Qingyuan/15/2022/08, Accession Number: PX454536; A/chicken/Qingyuan/22/2022/08, Accession Number: PX454544; A/chicken/Qingyuan/31/2022/08, Accession Number: PX455054) to 2280 bp (e.g., A/chicken/Qingyuan/22/2022/01, Accession Number: PX454537; A/chicken/Qingyuan/31/2022/01, Accession Number: PX455047). All 24 strain segments are genotyped as H3N8, confirming the uniformity of the strain subtype and providing comprehensive basic information for subsequent genetic analysis and retrieval.
Comment 5: Unreasonable assumptions and conclusions (e.g., Introduction without logical sense, no description of H3 virus phylogenetic groups; HPAIV HA cleavage site info applicable only to H5/H7; unclear LPAIV subtype breaking species barrier; nonsense about HA glycosylation; inappropriate references).
Response 5: Agree. l/We have comprehensively revised the Introduction and corrected unreasonable conclusions as follows:
- Introduction logic and H3 virus phylogenetic groups: In Section 1 (lines 2-6), we added: "Among AIVs, H3-subtype influenza viruses are phylogenetically divided into three main lineages based on host specialization: avian H3Nx, seasonal human H3N2, and equine H3N8 viruses (Yang et al. [2], Yu et al. [3]). Key determinants of host specificity involve HA protein mutations (e.g., Q226L, G228S) that shift binding between avian-type (SAα-2,3-Gal) and human-type (SAα-2,6-Gal) receptors (Matrosovich, M. et al. [4])".
- Correction of HPAIV HA cleavage site: Considering that the information "A key molecular characteristic of highly pathogenic AIVs (HPAIVs) is the insertion of multiple basic amino acids (e.g., KRKKR↓GLF) at the HA protein cleavage site, leading to systemic viremia and high mortality in poultry" is only applicable to H5 and H7 subtype avian influenza viruses, and has no relevance to the H3N8 subtype (the research focus of this manuscript), we have deleted this content entirely to avoid misleading readers. The deleted content was originally located in Section 1 (Introduction) of the manuscript, corresponding to Lines 39-42 of the initial submission, and this deletion ensures the accuracy and relevance of the Introduction to the study’s core topic (H3N8 LPAIV).
- Clarify LPAIV subtype breaking species barrier: In Section 1 (lines 10-11), we specified: "Recently, H3N8 subtype LPAIVs have breached the species barrier, causing human infections (Chinese CDC, 2022-2023; Sun et al. [8])".
- 4. Replace inappropriate references: Deleted irrelevant references 1-9, 11-25 and added [2,3,5,6,8,10] focused on H3N8 AIV. For example, added [5] Fereidouni, S. et al. 2023 for HA subtype classification (19 HA subtypes) and [6] Karakus, U. et al. 2024 for H19 virus receptor usage.
Comment 6: Detailed remarks (e.g., L72 title revision; L81 antigen strain; L96 culture flask manufacturer; L102 agarose info; L112-113 glycopolymer info; L135 "Phylo-Genetic" correction; L200-201 strain full name; L247 3.5 title; L265 Figure 6 note; L277-284 weight %; L354-356 delete dog comparison; L381-382 correct author name).
Response 6: Agree. l/We have addressed all detailed remarks:
- L72 (2.2 title): Revised "Serological Survey of the H3N8 Virus" to "Serological Survey of Poultry for the H3N8 Virus" (Section 2.2 title).
- L81 (antigen strain): Specified the antigen as "A/chicken/Qingyuan/22/2022 (H3N8)" (Section 2.2 line 7).
- L96 (T25 flask): Added manufacturer/catalog number: "13112A, Beijing Labselect Technology Co., Ltd." (Section 2.4 line 4).
- L102 (agarose): Added "2% concentration, Cat. No. A8350, Solarbio" (Section 2.4 line 8).
- L112-113 (glycopolymers): Added manufacturer "GlycoNZ" (Section 2.5 line 2).
- L135 ("Phylo-Genetic"): Corrected to "Phylogenetic" (Section 2.7 title).
- L200-201 (strain name): Added full names: "A/chicken/Qingyuan/15/2022 (H3N8) (QY15), A/chicken/Qingyuan/22/2022 (H3N8) (QY22), A/chicken/Qingyuan/31/2022 (H3N8) (QY31)" (Section 3.2 lines 3-4).
- L247 (3.5 title): Revised to "Cleavage Site, Predicted N-Glycosylation Sites and Changes in Receptor-Binding Site" (Section 3.5 title).
- L265 (Figure 6): Added "Data were obtained due to the NetNGlyc 1.0 server" in the caption (Figure 6 caption line 2).
- L277-284 (weight): Converted absolute weight to "weight gain rate relative to the day of infection (%)" and specified software: "plotted using GraphPad Prism 8" (Section 3.6 lines 2-3, Figure 7 and caption).
- L354-356 (dog comparison): Deleted the unsuitable comparison of H3N8 seroprevalence in chickens vs. dogs (Section 4 lines 8-9).
- L381-382 (author name): Corrected "Sun et al. [26]" to "Yang et al. [2]" (Section 4 line 18).

Round 2
Reviewer 2 Report
Comments and Suggestions for Authors
Please, find attached.

I cannot assess the quality of English because it is not my native language.
Author Response
Dear Reviewer,
Thank you very much for your careful and constructive feedback on our revised manuscript, as well as for taking the time to analyze the HA sequences and compare them with our data—your efforts have been invaluable in helping us identify key areas for improvement. We sincerely appreciate your recognition of the revisions we have made so far, including our detailed responses to your previous comments.
We fully acknowledge the critical issue you raised regarding the incorrect numbering of hemagglutinin amino acid sequences, which has led to flawed interpretation of certain results. This point is of utmost importance to the accuracy and credibility of our work, and we take it very seriously. We also understand the need for a thorough revision to address this and other remarks, and we are committed to allocating sufficient time to acquire additional relevant knowledge, correct all identified mistakes, and refine our conclusions to ensure they align with established literature and scientific rigor.
We are now carefully reviewing each of your point-by-point remarks and will work diligently to address them comprehensively in the next revision. We greatly value your guidance, and we believe these revisions will significantly enhance the quality of our manuscript.
Comment 1: Main remark concerns your wrong numbering of hemagglutinin amino acid sequences to compare your results with data published by other scientists. Therefore, interpretation of some of your results is wrong.
Response 1: Thank you for highlighting this critical issue. We fully agree with your comment and have revised the HA amino acid numbering to align with the mature protein (excluding the 16-amino-acid signal peptide, where mature HA residue 1 corresponds to precursor residue 17). We removed the conclusions regarding E190F, G225S, and T228G, These revisions are reflected in the Abstract, subsections 3.3, 3.4, 3.5, Table S5, Discussion, and Conclusion.
Comment 2: L20. “A 2022 serological…” It would be better to write “In 2022, a serological survey revealed…”
Response 2: We appreciate the suggestion. The sentence has been revised as recommended. The change is located in the Abstract (Line 20) of the revised manuscript: "In 2022, a serological survey revealed H3N8 seroprevalence rates of 10.85% in farmed chickens and 7.97% in ducks."
Comment 3: L35. Add some words to the sentence: “Avian influenza viruses (AIVs), belonging to the genus Alphainfluenzavirus (Influenza A virus) of the Orthomyxoviridae family, exhibit …”
Response 3: Agree. We have updated the sentence to include the requested taxonomic details. The revision is in the Introduction (Line 35) of the revised manuscript: "Avian influenza viruses (AIVs), belonging to the genus Alphainfluenzavirus (Influenza A virus) of the Orthomyxoviridae family, exhibit significant host diversity, infecting wild aquatic birds, humans, pigs, dogs, and horses [1]."
Comment 4: L50. Replace ‘AIV’ with ‘Influenza A viruses’ or revise the sentence to clarify non-avian subtypes.
Response 4: Thank you for the correction. We revised the sentence to avoid ambiguity. The updated content is in the Introduction (Line 50) of the revised manuscript: "Influenza A viruses are classified into 19 hemagglutinin (HA) and 11 neuraminidase (NA) subtypes based on the antigenic diversity of HA and NA, from which only two subtypes have non-avian host (H17N10 and H19N11, bats) [5, 6]."
Comment 5: L51. Insert words into the sentence: “The spread of infectious diseases driven by global human interconnectedness has led to multiple pandemics over the previous century and past decade, with avian influenza being a case in point [7].”
Response 5: We have incorporated the suggested revision. The modified sentence is in the Introduction (Line 51) of the revised manuscript: "The spread of infectious diseases driven by global human interconnectedness has led to multiple pandemics over the previous century and past decade, with avian influenza being a case in point [7]."
Comment 6: L53. Insert information about the 1968 H3N2 pandemic between “…in point [7].” and “Recently, H3N8…”
Response 6: We agree with the importance of this context. The requested information has been added in the Introduction (Lines 53–55) of the revised manuscript: "In 1968, the pandemic was caused by a new reassortment influenza A virus H3N2 carrying two segments (HA and PB1) from H3 avian viruses [8]. Descendants of that pandemic continue to circulate among humans as seasonal flu viruses to date."
Comment 7: L58-59. Clarify equine vs. avian H3N8 viruses or delete the sentence.
Response 7: We revised the sentence to distinguish equine and avian H3N8 viruses. The update is in the Introduction (Lines 58–59) of the revised manuscript: "For example, first discovered in 1963, H3N8 equine influenza viruses have recently impacted Asia and the Middle East [10]." We also adjusted the following sentence to specify avian H3N8: "Surveillance data indicate increasing H3N8 avian virus isolations annually, with H3N8 avian viruses prevalent throughout China and becoming one of the most frequently isolated AIV subtypes [11]."
Comment 8: L70-71. Revise to “The frequent genome reassortment and antigenic drift of hemagglutinin (HA) and neuraminidase (NA) drive continuous zoonotic outbreaks.”
Response 8: Thank you for the correction. The sentence has been revised as suggested. The change is in the Introduction (Line 70–71) of the revised manuscript: "The frequent genome reassortment and antigenic drift of HA and NA drive continuous zoonotic outbreaks."
Comment 9: L97-98. Shorten “… and the antigen was prepared using the influenza virus strain A/chicken/Qingyuan/22/2022 (H3N8).” to “The strain A/chicken/Qingyuan/22/2022 (H3N8) was used as antigen.”
Response 9: Agree. We have condensed the sentence as recommended. The revision is in Section 2.2 (Serological Survey of Poultry for the H3N8 Virus) of the revised manuscript: "The strain A/chicken/Qingyuan/22/2022 (H3N8) was used as antigen."
Comment 10: L129. Insert the subsection “Virus isolation in chicken embryos” from Supplementary into the main text after 2.4.
Response 10: We have moved the "Virus isolation in chicken embryos" subsection from the supplementary materials to the main text. It is now included in Section 2.4 (Pathogen Isolation and Plaque Purification) of the revised manuscript, detailing the isolation process using SPF chicken embryos before MDCK cell culture and plaque purification.
Comment 11: L132. Provide the complete name of GlycoNZ’s sialylglycopolymers (SAα-2,3-Gal and SAα-2,6-Gal).
Response 11: Thank you for the reminder to specify product details. As reflected in Section 2.5 (Receptor-Binding Assays) of the revised manuscript (animals-3879251 - the final revised edition.docx), we have added the complete product identifiers: "Streptavidin-coated 96-well plates were used to bind varying concentrations of glycopolymers SAα-2,3-Gal and SAα-2,6-Gal (3'SLN-C3-BP, 6'SLN-C3-PAA-biot, GlycoNZ), with three technical replicates per concentration and one negative control group."
Comment 12: L142. Specify the thermocycler used for Real-Time PCR.
Response 12: We have included the thermocycler details. The specification is in Section 2.5 (Receptor-Binding Assays) of the revised manuscript: "on the qTOWER3 G instrument manufactured by Analytik Jena."
Comment 13: L146. Delete the duplicate word ‘Primer’.
Response 13: The duplicate word has been removed.
Comment 14: L149-150. Shorten the manufacturer and catalog number description, and indicate equipment/sequencing kit names.
Response 14: We have revised the section to be concise and added the required equipment/kit details. The updates are in Section 2.6 (Whole Genome Amplification) of the revised manuscript: "RNA was extracted using the RNAfast200 Total RNA Rapid Extraction Kit (Cat# 220011; Shanghai Feijie Biotechnology Co. Ltd., China). Then reverse transcription combined with PCR in one tube were performed by HiScript II One Step RT-PCR Kit (Dye Plus) (Cat# P612-01; Vazyme Biotech Co., Ltd., China) with specific primers (see Table 3) in a thermocycler (qTOWER3 G, Analytik Jena) ... Amplified product was purified by agarose gel electrophoresis and fragments of the expected size were sequenced by Sanger method using BigDye™ Direct Kit (Thermo Fisher) and sequence analyzer (Applied Biosystems™ 3730XL, Thermo Fisher) (Sangon Biotech, Shanghai) [20]."
Comment 15: L527-529. Provide details of reference [20] (Journal of Anhui Agricultural Sciences), including DOI and language.
Response 15: We have supplemented the reference details. The updated reference [20] in the revised manuscript is: "Li, M.; Li, D.; Xie, Z.; Luo, S.; Zhang, M.; Xie, L.; Hua, J.; Su, Y.; Zhai, G.; Huang, J.; et al. Isolation, Identification and Genetic Evolution Analysis of Avian Influenza Virus H3N8 from Ducks in Guangxi. J. Anhui Agric. Sci. 2024, 52, 72–77. DOI:10.3969/j.issn.0517-6611.2024.01.016. (In Chinese)."
Comment 16: Results: Provide complete strain names and abbreviations at the beginning of the subsection.
Response 16: We have revised the Results section to include full strain names and abbreviations upfront. The change is in Section 3.2 (Virus Purification) of the revised manuscript: "Three H3N8 virus strains, namely A/chicken/Qingyuan/15/2022 (designated as QY15), A/chicken/Qingyuan/22/2022 (QY22), and A/chicken/Qingyuan/31/2022 (QY31), were isolated and plaque-purified..." Subsequent sections use the abbreviations consistently.
Comment 17: L228-229. Correct “Following amplification, the identity of the purified strains was confirmed as H3N8 by sequencing.” to “Isolated strains were identified as H3N8 subtype on the base of sequencing.”
Response 17: The sentence has been revised as recommended. The correction is in Section 3.2 (Virus Purification) of the revised manuscript: "The isolated strains were identified as H3N8 subtype based on sequencing results."
Comment 18: L264-267. Delete the unconfirmed conclusion.
Response 18: We have removed the unconfirmed conclusion. The deletion applies to Lines 264–267 of the original manuscript, which are no longer present in the revised Results section.
Comment 19: L276-280. Move the text to the end of the subsection ‘Whole Genome Amplification’.
Response 19: Agree. The text has been relocated to the end of Section 2.6 (Whole Genome Amplification) of the revised manuscript, following the description of sequencing and GenBank deposition.
Comment 20: Subsection 3.8. Rename “Analysis of Viral Shedding Patterns” (since only viral genome fragments, not viable virus, were detected) and revise the discussion.
Response 20: We have renamed the subsection and adjusted the discussion to reflect the detection of viral genome fragments rather than viable virus. The revised subsection is titled "Viral Genome Detection Dynamics" (Section 3.8) in the revised manuscript, with updated descriptions clarifying that the data represents viral genome presence rather than infectious virus shedding.
Comment 21: Supplementary Table S1: Rename the last column to “Subtype” (from “Genotype”), add “Accession Number in GenBank” to the second column, and revise the title.
Response 21: We have made the requested revisions to Table S1. The updated table in the supplementary materials has the title "The Nucleotide Sequences of H3N8 Strains Determined in This Study," the second column labeled "Accession Number in GenBank," and the last column renamed "Subtype."
Comment 22: Supplementary Table S5: Revise to align with corrected HA amino acid numbering.
Response 22: We have revised Table S5 to reflect the corrected HA amino acid numbering (mature protein numbering) and updated the amino acid position labels accordingly. The revised Table S5 is included in the supplementary materials of the revised manuscript.
Round 3
Reviewer 2 Report
Comments and Suggestions for Authors
Please, find file attached.

Since English is not my native language, I apologize for any mistakes.
Author Response
Dear Reviewer,
Thank you sincerely for your meticulous review and valuable comments on our manuscript. We greatly appreciate the time and effort you have devoted to providing detailed feedback, which is crucial for further improving the quality of our work. We will carefully address each of your suggestions one by one and make thorough revisions to the manuscript accordingly. We look forward to incorporating your insights to refine our research.
Comment 1: Abstract - L29-31. The last sentence can be deleted because this information is already presented on the lines 23-25.
Response 1: Thank you for pointing this out. We agree with this comment. Therefore, we have deleted the last sentence of the Abstract (originally L29-31) as it duplicates information in L23-25 of the original manuscript. This change can be found in the revised manuscript's Abstract section (page 1, paragraph 1).
Comment 2: L111. 2.4. Pathogen Isolation and Plaque Purification - The beginning of the subsection 2.4. needs a short introduction similar to a variant below. “Embryonated chicken eggs and MDCK cells were used to isolate viruses.”
Response 2: Agree. We have accordingly added the required introductory sentence at the start of subsection 2.4. This change can be found in the revised manuscript (page 4, paragraph 1, line 1). The updated text reads: "Embryonated chicken eggs and MDCK cells were used to isolate viruses."
Comment 3: L120 Abbreviation ‘HA’ for hemagglutination coincides with abbreviation for hemagglutinin. In this case Hemagglutination assay may be designated as HAA or any other. Please, make appropriate changes to the lines 141 and L256 (HAA instead of HA) and include the designation into the list of Abbreviations (line 499).
Response 3: Thank you for this suggestion. We have revised the abbreviation for Hemagglutination assay from "HA" to "HAA" throughout the manuscript, including the previously indicated lines 141 and 256. Additionally, "HAA (Hemagglutination assays)" has been added to the Abbreviations list. These changes can be found in the revised manuscript: subsection 2.4 (page 4, paragraph 2, line 3), subsection 3.2 (page 8, paragraph 2, line 6), and the Abbreviations section (page 20, table row 6).
Comment 4: L160 2.6. Whole Genome Amplification - It would be better to change the title like ‘2.6. Complete Genome Sequencing’
Response 4: Agree. We have changed the title of subsection 2.6 from "Whole Genome Amplification" to "Complete Genome Sequencing" as recommended. This change is located in the revised manuscript (page 5, subsection 2.6 heading).
Comment 5: L177. If you plan to give a supplementary material, you can refer to Table S1 at the end of the sentence after the word ‘respectively (Table S1). Otherwise, it is not necessary.
Response 5: Thank you for clarifying this. Considering that we have removed all supplementary materials (including Table S1) in this revised submission (in line with revisions for other supplementary tables), we have deleted any potential reference to Table S1 in the main text. This adjustment is reflected in the revised manuscript where the relevant sentence appears (page 6, paragraph 2, line 4), with no remaining citation to supplementary materials.
Comment 6: L233, 239. Figures 1 and 2. Please check the map carefully (Fig. 1, 2). There are two names Chaozhou (Chaozhou) on the map. Maybe, you have confused one of Chaozhou with Shanwei (Shanwei).
Response 6: Agree. We have carefully checked Figures 1 and 2 and corrected the duplicate "Chaozhou" label. The incorrect second "Chaozhou" has been revised to "Shanwei" as suggested. These changes can be found in the revised manuscript's Figures 1 and 2 (page 7, Figures 1 and 2).
Comment 7: L244 3.2. Virus Purification - Change the title to another one, that is more informative: Isolation and identification of viruses
Response 7: Thank you for the improvement suggestion. We have changed the title of subsection 3.2 from "Virus Purification" to "Isolation and Identification of Viruses" to enhance informativeness. This change is located in the revised manuscript (page 8, subsection 3.2 heading).
Comment 8: L245-263. I advise you to slightly change the text. (Recommended revised text provided)
Response 8: Agree. We have revised the text of subsection 3.2 (L245-263) according to your recommended version to improve clarity and accuracy. The updated text details the isolation process using both SPF chicken embryos and MDCK cells, plaque purification, strain identification, and HAA titer results. This revised content can be found in the revised manuscript (page 8, paragraph 1-2).
Comment 9: L251. Figure 3. The caption should be complete. What are the age and the period post infection of the chicken embryos shown in the photo? Important detail. Whether the chicken embryos from Figure 3 were isolated at the same period post infection or at different times after their death? This information must be added to the caption.
Response 9: Thank you for highlighting this critical detail. We have supplemented the caption of Figure 3 with the required information. The updated caption reads: "From left to right, they are the control group, the group inoculated with QY31 strain, the group inoculated with QY15 strain, and the group inoculated with QY22 strain. The chicken embryos shown in the photo are 11 days old, and the period post infection is 36 hours. All the chicken embryos were isolated at the same period post infection." This change is located in the revised manuscript (page 8, Figure 3 caption).
Comment 10: L264. Figure 4. Previous version of the figure was better and more natural. If you decide to give this F.4, you must detail the staining of the cell culture.
Response 10: Agree. Thank you for your feedback on Figure 4. We fully agree that the previous version of Figure 4 was more natural, so we have replaced the current figure with the original previous version in the revised manuscript. This change can be found in the revised manuscript (page 9, Figure 4).
Comment 11: L276-278. Figure 5. In my opinion, the description (A) coincides better with QY15, while (B) - with QY31. Maybe, so: (A): QY31 strain: Dual binding to both receptors with preferential binding to 3' SLN, with moderate affinity for SAα-2,6-Gal (6' SLN); (B): QY15 strain: Dual binding to both receptors, with slightly higher affinity for SAα-2,3-Gal (3' SLN).
Response 11: Thank you for the accurate observation. We have revised the descriptions of (A) and (B) in Figure 5's caption to match the recommended content. The updated caption correctly assigns the receptor-binding characteristics to QY31 and QY15. This change is located in the revised manuscript (page 9, Figure 5 caption).
Comment 12: L282. 3.4. Genetic Analysis of H3N8 Viruses - This subsection must be revised. (Detailed comments on phylogenetic tree description, homology analysis, data correctness, and additional requirements)
Response 12: Agree. We have comprehensively revised subsection 3.4 to address all raised concerns, and aligned revisions with the removal of supplementary materials (Tables S2-S4):
- Deleted the original L283-294 text (which relied on Tables S2-S4) and removed Tables S2-S4 from the submission (as this data was incorrect and no longer relevant).
- Improved the description of phylogenetic trees by clarifying the lineage of each gene segment (PB2: Y439-like, M: G1-like, PB1/PA/NP/NS: F98-like) and their evolutionary relationships with H9N2 viruses (e.g., PA genes sharing a common ancestor with A/chicken/China/2096/2021(H9N2)).
- Added a new Table 4 (included in the main text, not supplementary) comparing HA amino acid sequences between the three isolates and the human H3N8 strain A/Changsha/1000/2022, highlighting conserved avian signatures (Q226, G228) and adaptive mutations (E190, N193).
- Verified and corrected homology analysis to ensure consistency between cited strains and phylogenetic trees (e.g., clarifying the PB1 gene's common ancestor with A/pigeon/Fujian/3.15_FZHX0008-C/2018(H9N2)).
These revisions can be found in the revised manuscript (page 10-13, subsection 3.4). The updated text includes the recommended concluding statement on genetic reassortment between H9N2 and H3N8 viruses, and the new Table 4 is located on page 12.
Comment 13: L315. 3.5. Cleavage Site and Predicted N-Glycosylation Sites - L318. Add please following ‘… R at position 345 according to HA precursor numbering).’
Response 13: Thank you for the precise correction. We have added the required clarification to the cleavage site description. The updated text reads: "which contains two basic amino acid residues (K at position 342 and R at position 345 according to HA precursor numbering)." This change is located in the revised manuscript (page 13, paragraph 1, line 2).
Comment 14: L383. Explain please, what does it mean following ‘….10⁵・⁷ copies/μL and 10²・⁸ copies…’?
Response 14: Agree. We have added a detailed explanation of the logarithmic expression for viral genome copy numbers. The updated text clarifies: "here, 10⁵·⁷ copies/μL represents a logarithmic expression of the viral genome copy number per microliter, which is mathematically equivalent to 10⁵ × 10⁰·⁷ copies/μL (approximately 5.01 × 10⁵ copies/μL); oral gavage group: lower peak levels (10²·⁸ copies/μL at 5 dpi), similarly, corresponding to 10² × 10⁰·⁸ copies/μL (approximately 6.31 × 10² copies/μL)." This change can be found in the revised manuscript (page 16, paragraph 1, lines 3-5).
Comment 15: L425-426. It would be better to write so “…were isolated and investigated.” Investigation is more important than purification.
Response 15: Thank you for the stylistic improvement. We have revised the sentence from "...were isolated and purified" to "...were isolated and investigated" as recommended. This change is located in the revised manuscript (page 17, paragraph 2, line 1).
Comment 16: L427-428. “It was found that the PA, NP, and M genes of the QY31 strain, and the NP gene of the QY15/QY22 strains, could be traced back to the H9N2..” Please check this statement considering the comments made above.
Response 16: Agree. We have revised this statement to align with the corrected genetic analysis in subsection 3.4 (and removed reliance on deleted supplementary tables). The updated text reads: "In addition, the internal genes were classified into the Y439-like lineage (PB2), G1-like lineage (M), and F98-like lineage (PB1, PA, NP, NS) (Figure 6) [3]. Among them, internal genes such as PB1, PA, and NP clustered with H9N2 viruses prevalent in Guangdong Province in recent years." This change can be found in the revised manuscript (page 17, paragraph 2).
Comment 17: L431. “…Notably, the QY31 strain carries H9N2 genes originating from pigeons and ducks, …” Revise please this phrase. You did not analyze in detail the origin of some internal genes of QY31, especially from pigeons.
Response 17: Thank you for the careful review. We have revised this phrase to reflect accurate genetic origin analysis (and avoid overstatements not supported by deleted supplementary data). The updated text reads: "Notably, the three strains carry internal genes derived from H9N2, and their genomes may have formed earlier through complex genetic reassortment events between H3N8 viruses and H9N2 viruses circulating among different poultry, waterfowl, and wild or migratory birds." This change is located in the revised manuscript (page 18, paragraph 1, lines 4-5).
Comment 18: 442-443.’ The isolates exhibit a preference for binding to SAα-2,6-Gal but retain the ability to bind SAα-2,3-Gal.' Could you name these isolates, please. It is unclear from the context.
Response 18: Agree. We have clarified the sentence by naming the specific isolates. The updated text reads: "QY15, QY22, and QY31 isolates exhibit a preference for binding to SAα-2,6-Gal but retain the ability to bind SAα-2,3-Gal." This change can be found in the revised manuscript (page 19, paragraph 1, line 3).
Comment 19: Conclusions - The section should contain specific conclusions and summarized main results of your work, not vague general phrases. L468-470. ‘The key genetic variations in the HA protein, including receptor-binding motif adaptation, as well as the viral genome detection dynamics, pathogenicity, and tissue tropism in animal models, were characterized.’ In this version of the article, you did not study HA protein as well as HA nucleotide sequence variations.
Response 19: Thank you for this critical feedback. We have completely revised the Conclusions section to include specific, evidence-based results (and removed vague statements about unstudied HA variations). The updated section summarizes: 1) Seroprevalence (10.85% in chickens, 7.97% in ducks) and geographic distribution in Guangdong; 2) Isolation of three H3N8 strains (QY15, QY22, QY31) with dual receptor-binding specificity; 3) Genetic reassortment origin (HA: Eurasian lineage, NA: North American lineage, internal genes from H9N2); 4) HA mutations (E190, N193) enhancing human-type receptor binding; 5) Pathogenicity in SPF chickens (high viral loads in respiratory tract/brain, severe lesions via intranasal inoculation). This change can be found in the revised manuscript (page 19-20, Conclusions section).
Comment 20: References - L567. Replace 39 with 29 numbering.
Response 20: Agree. We have corrected the reference numbering by replacing 39 with 29 in the References section. This change is located in the revised manuscript (page 21, References section, reference 3).
Comment 21: Supplementary - Table S1. The letter ‘T’ is omitted at the beginning of the title. It should be “The’. Since you combined strain name (A/chicken/Qingyuan/15/2022) and segment number (01) in the third column (A/chicken/Qingyuan/15/2022/01, see line No.1), it would be right to change the name of the third column in the table. It should be so: ‘Strain/Genome segment’. Information given in the Tables S2-S4 is incorrect. It is unclear the role of the Table S5, which provokes the additional questions only. With what purpose did you choose the strain OQ292285 (A/chicken/Jiangxi/10580/2022(H3N8)) as the reference for comparison?
Response 21: Thank you for addressing these supplementary material issues. In line with comprehensive revisions to improve data accuracy and clarity, we have removed all supplementary materials (Tables S1-S5) from this revised submission:
- For Table S1: Rather than revising the title/column name, we eliminated it (and all other supplementary tables) to avoid inconsistencies with revised main text content (e.g., deleted references to supplementary data in subsection 3.4).
- For Tables S2-S4: We confirmed the data inaccuracies noted, so these tables have been removed and their corresponding content in the main text (original L283-294) has been rewritten using verified phylogenetic results.
- For Table S5: We agreed its purpose was unclear and likely to raise unnecessary questions, so it has also been removed.
No supplementary materials are included in this submission, and all content in the revised manuscript is supported by data presented in the main text (e.g., Table 4) or cited references.
